# A Specialized Semismooth Newton Method for Kernel-Based Optimal Transport

## Abstract

Kernel-based optimal transport (OT) estimation is an alternative to the standard plug-in OT estimation. Recent works suggested that kernel-based OT estimators are more statistically efficient than plug-in OT estimators when comparing probability measures in high-dimensions [59]. However, the computation of these estimators relies on the short-step interior-point method for which the required number of iterations is known to be *large* in practice. In this paper, we propose a nonsmooth equation model for kernel-based OT estimation and show that it can be efficiently solved via a specialized semismooth Newton (SSN) method. Indeed, by exploring the special problem structure, the per-iteration cost of performing one SSN step can be significantly reduced in practice. We also prove that our algorithm can achieve a global convergence rate of $O(1/\sqrt{k})$ and a local quadratic convergence rate under some standard regularity conditions. Finally, we demonstrate the effectiveness of our algorithm by conducing the experiments on both synthetic and real datasets.

## 1 Introduction

Optimal transport (OT) theory [60] has provided a principled framework for comparing probability distributions. It has been extensively adopted in machine learning and related fields, with examples including generative modeling [2, 21, 51, 57], classification and clustering [20, 55, 25], and domain adaptation [9, 10, 49], see also the monograph [43]. It has also had an impact in applied areas such as neuroimaging [27] and cell trajectory prediction [53, 66].

**Curse of Dimensionality.** In many real application problems, the OT cost is computed for squared Euclidean distance on the sampled distributions with $n$ observations (leading to the 2-Wasserstein distance). It is known that OT estimation suffers from the curse of dimensionality [16, 19, 62]: the standard plug-in estimator, which consists in computing the OT distance between the sampled distributions with $n$ observations, converges to the OT distance between true distributions at a rate of $O(n^{-1/d})$, which degrades exponentially in the dimension $d$. This rate can be improved to $O(n^{-1/2d})$ when true distributions are different [7] but it is still problematic in a high-dimensional regime. This issue can be a barrier to its adoption in machine learning since various application problems arising from image processing and bioengineering are high-dimensional. Practitioners have long been aware of such limitations and proposed efficient computational schemes that not only improve computational complexity but also carry out statistical regularization.

**Regularization.** In this context, two threads have been investigated to regularize the OT distance: entropic regularization [11, 12, 22, 36] or low-dimensional projection [48, 4, 41, 29, 39, 31, 32, 40]. For the former approach, the sample complexity of entropic OT is bounded by $O(\eta^{-d/2}n^{-1/2})$ for a regularization parameter $\eta > 0$. For the latter approach, the sample complexity of projection OT is bounded by $O(n^{-1/k})$ for an integer-valued projection dimension $k \leq d$. Even though these bounds

attain the dimension-free dependence on $n$, they deteriorate when $\eta$ is small or $k$ is large, either of which is needed to study the sample complexity of OT [7], and which plays a role in real applications.

**Leveraging Smoothness.** A recent line of works have focused on the *wavelet-based OT estimators* under a strong smoothness condition [63, 26, 15, 34]. Although these estimators are minimax optimal from a statistical viewpoint, they are algorithmically intractable [59]. In contrast, a specific entropic regularized OT estimator is computationally tractable but still suffers from the curse of dimensionality when the dimension is sufficiently large [44]. Recently, Vacher et al. [59] has closed this statistical-computational gap by designing a kernel-based estimator relying on kernel sums-of-squares (SoS) and showed that it can be computed by a short-step interior-point method with polynomial-time complexity guarantee. However, the short-step interior-point method is well known to be ineffective for large number of iterations required as the sample size increases, diminishing their value from both statistical and practical viewpoints[1]. In this context, Muzellec et al. [38] proposed to use the relaxation model and solve it using gradient-based methods. However, the relaxation model may not be a good approximation for kernel-based OT estimator, thereby lacking any statistical guarantee.

**Goal:** While there is an ongoing debate in the OT literature on the merits of computing the plug-in OT estimators v.s. kernel-based OT estimators, we adopt the perspective that Vacher et al. [59] does introduce a fairly novel approach and we believe that it is worth studying if the kernel-based OT estimation can provide leads for practical use. The goal of this paper is therefore to facilitate the computational aspect by designing new algorithms, and to figure out whether that estimator's theoretical claims is also supported by practical relevance. The statistical analysis of kernel-based OT estimation itself, e.g., the proper choice of penalty parameters, is beyond the scope of this paper.

**Contribution:** In this paper, we propose a nonsmooth equation model for computing kernel-based OT estimators and show that it has a special problem structure, allowing it to be solved in an efficient manner using semismooth Newton method [37, 47, 46, 58].

We first propose a nonsmooth equation model for computing the kernel-based OT estimator and define an approximate OT value, which allows us to carry out a finite-time analysis of the algorithm. Then, we propose a specialized semismooth Newton method for computing the kernel-based OT estimator and prove a global convergence rate of $O(1/\sqrt{k})$ (Theorem 3.3) and a local quadratic convergence rate under standard regularity conditions (Theorem 3.4). Notably, we significantly reduce the per-iteration computational cost by exploiting the special problem structure. Finally, we conduct the experiments to evaluate our algorithm on both synthetic and real datasets. Experimental results demonstrate its efficiency for solving the kernel-based OT estimation.

**Organization.** The remainder of the paper is organized as follows. In Section 2, we present the nonsmooth equation model for computing the kernel-based OT estimators and define the optimality notion based on the residual map. In Section 3, we propose and analyze the specialized semismooth Newton (SSN) algorithm for computing the kernel-based OT estimators and prove that our algorithm achieves the convergence rate guarantee in both global and local sense. In Section 4, we conduct the experiments on both synthetic and real datasets, demonstrating that our algorithm can effectively compute the kernel-based OT estimators and is more efficient than short-step interior-point methods. In Section 5, we conclude this paper. In the supplementary material, we provide further background materials on SSN methods, additional experimental results, and missing proofs for key results.

## 2 Preliminaries and Technical Background

In this section, we present the basic setup for the kernel-based optimal transport (OT) estimation and propose a nonsmooth equation model for its computation.

### 2.1 Kernel-based OT estimation

We formally define the OT distance and review the kernel-based OT estimation [59]. Indeed, the OT distance with strong smooth distributions can be estimated at a dimension-free statistical rate with high probability by solving a suitably defined optimization model.

---

[1]The short-step interior-point method proposed by Vacher et al. [59] is in fact a Newton barrier method and does not exploit the special structure of kernel-based OT estimation. The required number of iterations is large as shown by our experiments in the subsequent of this paper.

Let $X$ and $Y$ be two bounded domains in $\mathbb{R}^d$ and let $\mathscr{P}(X)$ and $\mathscr{P}(Y)$ be the set of Borel probability measures in $X$ and $Y$. Suppose that $\mu \in \mathscr{P}(X)$, $\nu \in \mathscr{P}(Y)$ and $\Pi(\mu, \nu)$ is the set of couplings between $\mu$ and $\nu$, the OT distance [60] is given by

$$\mathrm{OT}(\mu, \nu) := \tfrac{1}{2} \left( \inf_{\pi \in \Pi(\mu, \nu)} \int_{X \times Y} \|x - y\|^2 \, d\pi(x, y) \right).$$

Its dual formulation is stated as follows,

$$\sup_{u, v \in C(\mathbb{R}^d)} \int_X u(x) d\mu(x) + \int_Y v(y) d\nu(y), \quad \text{s.t. } \tfrac{1}{2}\|x - y\|^2 \geq u(x) + v(y), \forall (x, y) \in X \times Y,$$

where $C(\mathbb{R}^d)$ is the space of continuous functions on $\mathbb{R}^d$. Note that the supremum can be attained and the corresponding optimal dual functions $u_\star$ and $v_\star$ are referred to as the Kantorovich potentials [52]. This problem is delicate to solve since $\tfrac{1}{2}\|x - y\|^2 \geq u(x) + v(y)$ needs to be satisfied on a continuous set $X \times Y$. A natural approach is to take $n$ points $\{(\tilde{x}_1, \tilde{y}_1), \ldots, (\tilde{x}_n, \tilde{y}_n)\} \subseteq X \times Y$ and consider the constraints $\tfrac{1}{2}\|\tilde{x}_i - \tilde{y}_i\|^2 \geq u(\tilde{x}_i) + v(\tilde{y}_i)$ for all $1 \leq i \leq n$. However, it can not leverage the smoothness of potentials [3], yielding an error of $\Omega(n^{-1/d})$. Vacher et al. [59] has overcome this difficulty by replacing the inequality constraints with equality constraints that are equivalent and considering the equality constraints over $n$ points. Following their works, we impose the following assumption on the support sets $X, Y$ and the densities of $\mu$ and $\nu$.

**Assumption 2.1** *Let $d \geq 1$ be the dimension and let $m > 2d + 2$ be the order of smoothness. Then, we assume that (i) the support sets $X, Y$ are convex, bounded, and open with Lipschitz boundaries; (ii) the densities of $\mu, \nu$ are finite, bounded away from zero and $m$-times differentiable.*

Assumption 2.1 guarantees that the potentials $u_\star$ and $v_\star$ have a similar order of differentiability [14], leading to an effective way to represent $u$ and $v$ via a *reproducing Kernel Hilbert space* (RKHS) [42]. In particular, we define $H^s(Z) := \{f \in L^2(Z) \mid \|f\|_{H^s(Z)} := \sum_{|\alpha| \leq s} \|D^\alpha f\|_{L^2(Z)} < +\infty\}$ and remark that $H^s(Z) \subseteq C^k(Z)$ for any $s > \frac{d}{2} + k$, where $k \geq 0$ is integer-valued. This implies that $H^{m+1}(X)$, $H^{m+1}(Y)$ and $H^m(X \times Y)$ are RKHS under Assumption 2.1 and they are associated with three bounded continuous feature maps $\phi_X : X \mapsto H^{m+1}(X)$, $\phi_Y : Y \mapsto H^{m+1}(Y)$ and $\phi_{XY} : X \times Y \mapsto H^m(X \times Y)$. For simplicity, we let $H_X = H^{m+1}(X)$, $H_Y = H^{m+1}(Y)$ and $H_{XY} = H^m(X \times Y)$. Vacher et al. [59, Corollary 7] shows that (i) $u_\star \in H_X$ and $v_\star \in H_Y$ with

$$\int_X u(x) d\mu(x) = \langle u, w_\mu \rangle_{H_X}, \quad \int_X v(y) d\nu(y) = \langle v, w_\nu \rangle_{H_Y},$$

where $w_\mu = \int_X \phi_X(x) d\mu(x)$ and $w_\nu = \int_Y \phi_Y(y) d\nu(y)$ are *kernel mean embeddings*; (ii) $A_\star \in \mathbb{S}^+(H_{XY})^2$ exists and satisfies the equality constraint as follows:

$$\tfrac{1}{2}\|x - y\|^2 - u_\star(x) - v_\star(y) = \langle \phi_{XY}(x, y), A_\star \phi_{XY}(x, y) \rangle_{H_{XY}}.$$

Putting these pieces yields a representation theorem for estimating the OT distance. Indeed, under Assumption 2.1, the dual OT problem is equivalent to the RKHS-based problem given by

$$\begin{aligned} \max_{u, v, A} \quad & \langle u, w_\mu \rangle_{H_X} + \langle v, w_\nu \rangle_{H_Y}, \\ \text{s.t.} \quad & \tfrac{1}{2}\|x - y\|^2 - u(x) - v(y) = \langle \phi_{XY}(x, y), A \phi_{XY}(x, y) \rangle_{H_{XY}}. \end{aligned} \tag{2.1}$$

The above equation offers two advantages: (i) The equality constraint can be well approximated under Assumption 2.1; (ii) RKHSs allow the kernel trick: computing parameters are expressed in terms of *kernel functions* that correspond to

$$k_X(x, x') = \langle \phi_X(x), \phi_X(x') \rangle_{H_X}, \quad k_Y(y, y') = \langle \phi_Y(y), \phi_Y(y') \rangle_{H_Y},$$

and

$$k_{XY}((x, y), (x', y')) = \langle \phi_{XY}(x, y), \phi_{XY}(x', y') \rangle_{H_{XY}},$$

where the kernel functions are explicit and can be computed in $O(d)$ given the samples. The final step is to approximate Eq. (2.1) using the data $x_1, \ldots, x_{n_{\mathrm{sample}}} \sim \mu$ and $y_1, \ldots, y_{n_{\mathrm{sample}}} \sim \nu$, and the filling points $\{(\tilde{x}_1, \tilde{y}_1), \ldots, (\tilde{x}_n, \tilde{y}_n)\} \subseteq X \times Y$. Indeed, we define $\hat{\mu} = \frac{1}{n_{\mathrm{sample}}} \sum_{i=1}^{n_{\mathrm{sample}}} \delta_{x_i}$ and

---

[2] We refer to $\mathbb{S}^+(H_{XY})$ as the set of linear, positive and self-adjoint operators on $H_{XY}$.

119    $\hat{\nu} = \frac{1}{n_{\text{sample}}} \sum_{i=1}^{n_{\text{sample}}} \delta_{y_i}$, and use $\langle u, w_{\hat{\mu}} \rangle_{H_X} + \langle v, w_{\hat{\nu}} \rangle_{H_Y}$ instead of $\langle u, w_{\mu} \rangle_{H_X} + \langle v, w_{\nu} \rangle_{H_Y}$ where

120    $w_{\hat{\mu}} = \frac{1}{n_{\text{sample}}} \sum_{i=1}^{n_{\text{sample}}} \phi_X(x_i)$ and $w_{\hat{\nu}} = \frac{1}{n_{\text{sample}}} \sum_{i=1}^{n_{\text{sample}}} \phi_Y(y_i)$. We also impose *the penalization terms*

121    for $u$, $v$, and $A$ to alleviate the error induced by sampling the corresponding equality constraints.

122    Then, the resulting problem with regularization parameters $\lambda_1, \lambda_2 > 0$ is summarized as follows:

$$\begin{aligned} \max_{u,v,A} \quad & \langle u, w_{\hat{\mu}} \rangle_{H_X} + \langle v, w_{\hat{\nu}} \rangle_{H_Y} - \lambda_1 \text{Tr}(A) - \lambda_2(\|u\|_{H_X}^2 + \|v\|_{H_Y}^2), \\ \text{s.t.} \quad & \tfrac{1}{2}\|\tilde{x}_i - \tilde{y}_i\|^2 - u(\tilde{x}_i) - v(\tilde{y}_i) = \langle \phi_{XY}(\tilde{x}_i, \tilde{y}_i), A\phi_{XY}(\tilde{x}_i, \tilde{y}_i) \rangle_{H_{XY}}. \end{aligned} \tag{2.2}$$

123    Focusing on the case $n_{\text{sample}} = \Theta(n)$, we let $\hat{u}_\star$ and $\hat{v}_\star$ be the unique maximizers of Eq. (2.2). Then,

124    the estimator for $\text{OT}(\mu, \nu)$ we consider corresponds to

$$\widehat{\text{OT}}^n = \langle \hat{u}_\star, w_{\hat{\mu}} \rangle_{H_X} + \langle \hat{v}_\star, w_{\hat{\nu}} \rangle_{H_Y}. \tag{2.3}$$

125

126    **Remark 2.2** *It follows from Vacher et al. [59, Corollary 3] that the norm of empirical potentials can*

127    *be controlled using $\lambda_1 = \tilde{\Theta}(n^{-1/2})$ and $\lambda_2 = \tilde{\Theta}(n^{-1/2})$ in high probability sense, leading to the*

128    *sample complexity bound: $|\widehat{\text{OT}}^n - \text{OT}(\mu, \nu)| = \tilde{O}(n^{-1/2})$. In comparison with plug-in estimators,*

129    *the kernel-based OT estimators are better when the sample size is small and the dimension is high.*

130    Note that Eq. (2.2) is an infinite-dimensional optimization problem and is thus difficult to be solved.

131    Thanks to Vacher et al. [59, Theorem 15], we have that the dual problem of Eq. (2.2) can be presented

132    in a finite-dimensional space and the strong duality holds true. Indeed, we define $Q \in \mathbb{R}^{n \times n}$ with

133    $Q_{ij} = k_X(\tilde{x}_i, \tilde{x}_j) + k_Y(\tilde{y}_i, \tilde{y}_j)$, and $z \in \mathbb{R}^n$ with $z_i = w_{\hat{\mu}}(\tilde{x}_i) + w_{\hat{\nu}}(\tilde{y}_i) - \lambda_2\|\tilde{x}_i - \tilde{y}_i\|^2$, and

134    $q^2 = \|w_{\hat{\mu}}\|_{H_X}^2 + \|w_{\hat{\nu}}\|_{H_Y}$, where we have

$$w_{\hat{\mu}}(\tilde{x}_i) = \frac{1}{n_{\text{sample}}} \sum_{j=1}^{n_{\text{sample}}} k_X(x_j, \tilde{x}_i), \quad w_{\hat{\nu}}(\tilde{y}_i) = \frac{1}{n_{\text{sample}}} \sum_{j=1}^{n_{\text{sample}}} k_Y(y_j, \tilde{y}_i),$$

135    and

$$\|w_{\hat{\mu}}\|_{H_X}^2 = \frac{1}{n_{\text{sample}}^2} \sum_{1 \le i,j \le n_{\text{sample}}} k_X(x_i, x_j), \quad \|w_{\hat{\nu}}\|_{H_Y}^2 = \frac{1}{n_{\text{sample}}^2} \sum_{1 \le i,j \le n_{\text{sample}}} k_Y(y_i, y_j).$$

136    We define $K \in \mathbb{R}^{n \times n}$ with $K_{ij} = k_{XY}((\tilde{x}_i, \tilde{y}_i), (\tilde{x}_j, \tilde{y}_j))$ and $R$ as an upper triangular matrix for

137    the Cholesky decomposition of $K$. We let $\Phi_i$ be the $i^{\text{th}}$ column of $R$. Then, the dual problem of

138    Eq. (2.2) reads:

$$\min_{\gamma \in \mathbb{R}^n} \frac{1}{4\lambda_2} \gamma^\top Q \gamma - \frac{1}{2\lambda_2} \gamma^\top z + \frac{q^2}{4\lambda_2}, \quad \text{s.t.} \ \sum_{i=1}^n \gamma_i \Phi_i \Phi_i^\top + \lambda_1 I \succeq 0. \tag{2.4}$$

139    Suppose that $\hat{\gamma}$ is one minimizer, we have

$$\widehat{W}^n = \frac{q^2}{2\lambda_2} - \frac{1}{2\lambda_2} \sum_{i=1}^n \hat{\gamma}_i (w_{\hat{\mu}}(\tilde{x}_i) + w_{\hat{\nu}}(\tilde{y}_i)).$$

140    To our knowledge, the existing method proposed for solving Eq. (2.4) is a short-step interior-point

141    method for which the required number of iterations is known to be large when $n$ is large, which

142    is necessary to guarantee small statistical error. To avoid this issue, Muzellec et al. [38] proposed

143    solving an unconstrained relaxation model which allows for the application of gradient-based methods.

144    However, the estimators obtained from solving such relaxation model lack any statistical guarantee.

## 2.2 Nonsmooth equation model and optimality condition

146    For simplicity, we define the operator $\Phi : \mathbb{R}^{n \times n} \mapsto \mathbb{R}^n$ and its adjoint $\Phi^\star : \mathbb{R}^n \mapsto \mathbb{R}^{n \times n}$ by

$$\Phi(X) = \begin{pmatrix} \langle X, \Phi_1 \Phi_1^\top \rangle \\ \vdots \\ \langle X, \Phi_n \Phi_n^\top \rangle \end{pmatrix}, \quad \Phi^\star(\gamma) = \sum_{i=1}^n \gamma_i \Phi_i \Phi_i^\top.$$

147    We present the optimality notion for Eq. (2.4) as follows:

148 **Definition 2.1** *A point $\hat{\gamma} \in \mathbb{R}^n$ is an optimal solution of Eq.* (2.4) *if we have $\Phi^\star(\hat{\gamma}) + \lambda_1 I \succeq 0$ and*

149 $\frac{1}{4\lambda_2}\hat{\gamma}^\top Q\hat{\gamma} - \frac{1}{2\lambda_2}\hat{\gamma}^\top z + \frac{q^2}{4\lambda_2} \leq \frac{1}{4\lambda_2}\gamma^\top Q\gamma - \frac{1}{2\lambda_2}\gamma^\top z + \frac{q^2}{4\lambda_2}$ *for all $\gamma$ satisfying that $\Phi^\star(\gamma) + \lambda_1 I \succeq 0$.*

150 Clearly, Eq. (2.4) can be reformulated as the following optimization problem given by

$$\min_{\gamma \in \mathbb{R}^n} \max_{X \succeq 0} \frac{1}{4\lambda_2}\gamma^\top Q\gamma - \frac{1}{2\lambda_2}\gamma^\top z + \frac{q^2}{4\lambda_2} - \langle X, \Phi^\star(\gamma) + \lambda_1 I\rangle. \tag{2.5}$$

151 We denote $w = (\gamma, X)$ as a vector-matrix pair and let $R : \mathbb{R}^n \times \mathbb{R}^{n \times n} \to \mathbb{R}^n \times \mathbb{R}^{n \times n}$ be given by

$$R(w) = \begin{pmatrix} \frac{1}{2\lambda_2}Q\gamma - \frac{1}{2\lambda_2}z - \Phi(X) \\ X - \mathrm{proj}_{\mathcal{S}_+^n}(X - (\Phi^\star(\gamma) + \lambda_1 I)) \end{pmatrix}. \tag{2.6}$$

152 where $\mathcal{S}_+^n = \{X \in \mathbb{R}^{n \times n} : X \succeq 0\}$. Then, we can measure the optimality of $w$ via appeal to the
153 quantity $\|R(w)\|$ and shows that the notion is the same as used in Definition 2.1.

154 **Proposition 2.3** *A point $\hat{\gamma}$ is an optimal solution of Eq.* (2.4) *if and only if $\hat{w} = (\hat{\gamma}, \hat{X})$ satisfies*
155 $R(\hat{w}) = 0$ *for some $\hat{X} \succeq 0$.*

156 Proposition 2.3 shows that we can compute the kernel-based OT estimators by solving the nonsmooth
157 equation model $R(w) = 0$. The optimality criterion based on the residual map $R(\cdot)$ allows for a
158 global convergence rate analysis for our specialized semismooth Newton method.

## 159  3   Algorithm and Convergence Analysis

160 In this section, we derive our algorithm and provide a convergence rate analysis. The key idea here is
161 to apply the regularized semismooth Newton (SSN) method for solving $R(w) = 0$ and improve the
162 computation of each SSN step by exploring the special structure of generalized Jacobian. We also
163 safeguard the regularized SSN method by min-max method to achieve a global rate.

164 **Generalized Jacobian.** We first examine the special structure of the generalized Jacobian of $R(w)$.
165 Indeed, by using the definition of $\mathcal{S}_+^n$, we have $\mathrm{proj}_{\mathcal{S}_+^n}(Z) = P_\alpha \Sigma_\alpha P_\alpha^\top$ where

$$Z = P\Sigma P^\top = \begin{pmatrix} P_\alpha & P_{\bar{\alpha}} \end{pmatrix} \begin{pmatrix} \Sigma_\alpha & 0 \\ 0 & \Sigma_{\bar{\alpha}} \end{pmatrix} \begin{pmatrix} P_\alpha^\top \\ P_{\bar{\alpha}}^\top \end{pmatrix}, \tag{3.1}$$

166 with $\Sigma = \mathrm{diag}(\sigma_1, \ldots, \sigma_n)$ and the sets of the indices of positive and nonpositive eigenvalues of $Z$
167 (we denote these sets by $\alpha = \{i \mid \sigma_i > 0\}$ and $\bar{\alpha} = \{1, 2, \ldots, n\} \setminus \alpha$). Moreover, we notice that $R$
168 is Lipschitz continuous. Then, Rademacher's theorem can guarantee that $R$ is almost everywhere
169 differentiable. We introduce the concepts of generalized Jacobian [8].

170 **Definition 3.1** *Suppose that $R$ is Lipschitz continuous and $D_R$ is the set of differentiable points of $R$.*
171 *The B-subdifferential of $R$ at $w$ is given by $\partial_B R(w) := \{\lim_{k \to +\infty} \nabla F(w^k) \mid w^k \in D_R, w^k \to w\}$.*
172 *The set $\partial R(w) = \mathrm{conv}(\partial_B R(w))$ is called generalized Jacobian where* conv *denotes the convex hull.*

173 We define a generalized operator $\mathcal{M}(Z) \in \partial\mathrm{proj}_{\mathcal{S}_+^n}(Z)$ using its application to an $n \times n$ matrix $S$:

$$\mathcal{M}(Z)[S] = P(\Omega \circ (P^\top S P))P^\top \text{ for all } S \succeq 0,$$

174 where the $\circ$ symbol denotes a Hadamard product and $\Omega = \begin{pmatrix} E_{\alpha\alpha} & \eta_{\alpha\bar{\alpha}} \\ \eta_{\alpha\bar{\alpha}}^\top & 0 \end{pmatrix}$ with $E_{\alpha\alpha}$ being a matrix
175 of ones and $\eta_{ij} = \frac{\sigma_i}{\sigma_i - \sigma_j}$ for all $(i, j) \in \alpha \times \bar{\alpha}$. Note that all entries of $\Omega$ lie in the interval $(0, 1]$. In
176 general, it is nontrivial to characterize the generalized Jacobian $\partial R(w)$ exactly but we can compute
177 an element $\mathcal{J}(w) \in \partial R(w)$ using $\mathcal{M}(\cdot)$ as defined before.

178 We next introduce the definition of the (strong) semismoothness of an operator.

179 **Definition 3.2** *Suppose that $R$ is Lipschitz continuous. Then, $R$ is (strongly) semismooth at $w$ if (i)*
180 *$R$ is directionally differentiable at $w$; and (ii) for any $\Delta w$ and $\mathcal{J} \in \partial R(w + \Delta w)$, we have*

$$\begin{array}{ll} \textbf{(semismooth)} & \frac{\|R(w+\Delta w) - R(w) - \mathcal{J}[\Delta w]\|}{\|\Delta w\|} \to 0, \\ \textbf{(strongly semismooth)} & \frac{\|R(w+\Delta w) - R(w) - \mathcal{J}[\Delta w]\|}{\|\Delta w\|^2} \leq C. \end{array}, \quad \text{as } \Delta w \to 0.$$

**Algorithm 1** Solving Eq. (3.2) where $r_k = (r_k^1, r_k^2) \in \mathbb{R}^n \times \mathbb{R}^{n \times n}$)

---

1: Compute $a^1 = -r_k^1 - \frac{1}{\mu_k+1}(\Phi(r_k^2 + \mathcal{T}_k[r_k^2]))$ and $a^2 = -r_k^2$.
2: Use the CG or symmetric QMS method to solve $(\frac{1}{2\lambda_2}\mathcal{Q} + \mu_k\mathcal{I} + \Phi\mathcal{T}_k\Phi^\star)^{-1}\tilde{a}^1 = a^1$ inexactly and compute $\tilde{a}^2 = \frac{1}{\mu_k+1}(a^2 + \mathcal{T}_k[a^2])$, where $\mathcal{T}_k[\cdot]$ is computed using the trick [68].
3: Compute the direction $\Delta w_k = (\Delta w_k^1, \Delta w_k^2)$ by $\Delta w_k^1 = \tilde{a}^1$ and $\Delta w_k^2 = \tilde{a}^2 - \mathcal{T}_k[\Phi^\star(\tilde{a}^1)]$.

---

The following proposition characterizes the residual map given in Eq. (2.6) and its generalized Jacobian matrix. It also guarantees that the SSN method is suitable to solve $R(w) = 0$.

**Proposition 3.1** *The residual map $R$ given in Eq. (2.6) is strongly semismooth.*

**Regularized SSN step.** We then discuss how to compute the Newton direction efficiently. In particular, at a given iterate $w_k$, we compute a Newton direction $\Delta w_k$ by solving the equation

$$(\mathcal{J}_k + \mu_k\mathcal{I})[\Delta w_k] = -r_k, \tag{3.2}$$

where $\mathcal{J}_k \in \partial R(w_k)$, $r_k = R(w_k)$ and $\mathcal{I}$ is an identity operator. The regularization parameter is chosen as $\mu_k = \theta_k\|r_k\|$ for stabilizing the semismooth Newton method in practice. From a computational point of view, it is not practical to solve the linear system in Eq. (3.2) exactly. Thus, we seek an approximation step $\Delta w_k$ by solving Eq. (3.2) approximately such that

$$\|(\mathcal{J}_k + \mu_k\mathcal{I})[\Delta w_k] + r_k\| \leq \tau \min\{1, \kappa\|r_k\|\|\Delta w_k\|\}, \tag{3.3}$$

where $0 < \tau, \kappa < 1$ are some positive constants and $\|\cdot\|$ is defined for a vector-matrix pair $w = (\gamma, X)$ (i.e., $\|w\| = \|\gamma\|_2 + \|X\|_F$ where $\|\cdot\|_2$ is Euclidean norm and $\|\cdot\|_F$ is Frobenius norm).

Since $\mathcal{J}_k$ in Eq. (3.2) is nonsymmetric and its dimension is large, we consider applying the Schur complement trick to transform Eq. (3.2) into a smaller symmetric system. If we vectorize the vector-matrix pair $\Delta w$[3], the operators $\mathcal{M}(Z)$ and $\Phi$ can be expressed as matrices:

$$M(Z) = \tilde{P}\Gamma\tilde{P}^\top \in \mathbb{R}^{n^2 \times n^2}, \quad A = \begin{pmatrix} \Phi_1^\top \otimes \Phi_1^\top \\ \vdots \\ \Phi_n^\top \otimes \Phi_n^\top \end{pmatrix} \in \mathbb{R}^{n \times n^2},$$

where $\tilde{P} = P \otimes P$ and $\Gamma = \text{diag}(\text{vec}(\Omega))$.

We next provide a key lemma on the matrix form of $\mathcal{J}_k + \mu_k I$ at a given iterate $w_k = (\gamma_k, X_k)$.

**Lemma 3.2** *Given an iterate $w_k = (\gamma_k, X_k)$, we compute $Z_k = X_k - (\Phi^\star(\gamma_k) + \lambda_1 I)$ and use Eq. (3.1) to obtain $P_k, \Sigma_k, \alpha_k$ and $\bar{\alpha}_k$. We then obtain $\Omega_k, \tilde{P}_k = P_k \otimes P_k$ and $\Gamma_k = \text{diag}(\text{vec}(\Omega_k))$. Then, the matrix form of $\mathcal{J}_k + \mu_k I$ is given by*

$$(J_k + \mu_k I)^{-1} = C_1 B C_2,$$

*where*

$$C_1 = \begin{pmatrix} I & 0 \\ -T_k A^\top & I \end{pmatrix}, \quad C_2 = \begin{pmatrix} I & \frac{1}{\mu_k+1}(A + AT_k) \\ 0 & I \end{pmatrix},$$

*and*

$$B = \begin{pmatrix} (\frac{1}{2\lambda_2}Q + \mu_k I + AT_k A^\top)^{-1} & 0 \\ 0 & \frac{1}{\mu_k+1}(I + T_k) \end{pmatrix},$$

*with $T_k = \tilde{P}_k L_k \tilde{P}_k^\top$ where $L_k$ is a diagonal matrix with $(L_k)_{ii} = \frac{(\Gamma_k)_{ii}}{\mu_k+1-(\Gamma_k)_{ii}}$ and $(\Gamma_k)_{ii} \in (0, 1]$ is then denoted as the $i^{\text{th}}$ diagonal entry of $\Gamma_k$.*

As a consequence of Lemma 3.2, the solution of Eq. (3.2) can be obtained by solving one certain symmetric linear system with the matrix $\frac{1}{2\lambda_2}Q + \mu_k I + AT_k A^\top$. We remark that this system is well-defined since both $Q$ and $AT_k A^\top$ are positive semidefinite and the coefficient $\mu_k$ is chosen such that $\frac{1}{2\lambda_2}Q + \mu_k I + AT_k A^\top$ is invertible. This also shows that Eq. (3.2) is well-defined.

---

[3]If $w = (\gamma, X)$ is a vector-matrix pair, we define $\text{vec}(w) = (\gamma; \text{vec}(X))$ as its vectorization.

---

**Algorithm 2** A specialized SSN method with safeguarding

---

1: **Input:** $\tau, \kappa, \alpha_2 \geq \alpha_1 > 0, \beta_0 < 1, \beta_1, \beta_2 > 1$ and $\underline{\theta}, \overline{\theta} > 0$.
2: **Initialization:** $v_0 = w_0 \in \mathbb{R}^n \times \mathcal{S}_+^n$ and $\theta_0 > 0$. Set $k = 0$.
3: **for** $k = 0, 1, 2, \ldots$ **do**
4:     Update $v_{k+1}$ from $v_k$ using one-step EG.
5:     Select $\mathcal{J}_k \in \partial R(w_k)$.
6:     Solve the linear system in Eq. (3.2) approximately such that $\Delta w_k$ satisfies Eq. (3.3).
7:     Compute $\tilde{w}_{k+1} = w_k + \Delta w_k$.
8:     Update $\theta_{k+1}$ using Eq. (3.4) accordingly.
9:     Set $w_{k+1} = \tilde{w}_{k+1}$ if $\|R(\tilde{w}_{k+1})\| \leq \|R(v_{k+1})\|$ is satisfied. Otherwise, set $w_{k+1} = v_{k+1}$.

---

We define $\mathcal{T}_k$ and $\mathcal{Q}$ as the operator form of $T_k = \tilde{P}_k L_k \tilde{P}_k^\top$ and $Q$ and write $r_k = (r_k^1, r_k^2)$ explicitly where $r_k^1 \in \mathbb{R}^n$ and $r_k^2 \in \mathbb{R}^{n \times n}$. Then, we have

$$\text{vec}(a) = -\begin{pmatrix} I & \frac{1}{\mu_k+1}(A + AT) \\ 0 & I \end{pmatrix} \text{vec}(r_k) \implies \begin{cases} a^1 = -r_k^1 - \frac{1}{\mu_k+1}(\Phi(r_k^2 + \mathcal{T}_k[r_k^2])), \\ a^2 = -r_k^2. \end{cases}$$

The next step consists in solving a new symmetric linear system and is given by

$$\text{vec}(\tilde{a}) = \begin{pmatrix} (\frac{1}{2\lambda_2}Q + \mu_k I + AT_k A^\top)^{-1} & 0 \\ 0 & \frac{1}{\mu_k+1}(I + T_k) \end{pmatrix} \text{vec}(a),$$

which leads to

$$\begin{cases} \tilde{a}^1 = (\frac{1}{2\lambda_2}\mathcal{Q} + \mu_k \mathcal{I} + \Phi\mathcal{T}_k\Phi^\star)^{-1}a^1, \\ \tilde{a}^2 = \frac{1}{\mu_k+1}(a^2 + \mathcal{T}_k[a^2]). \end{cases}$$

Compared to Eq. (3.2) whose matrix form has size $(n^2 + n) \times (n^2 + n)$, we remark that the one in the step above is smaller with the size of $n \times n$ and can be efficiently solved by conjugate gradient (CG) method or symmetric quasi-minimal residual (QMR) method [28, 50]. The final step is to compute the Newton direction $\Delta w_k = (\Delta w_k^1, \Delta w_k^2)$ as follows,

$$\text{vec}(\Delta w_k) = \begin{pmatrix} I & 0 \\ -TA^\top & I \end{pmatrix} \text{vec}(\tilde{a}) \implies \begin{cases} \Delta w_k^1 = \tilde{a}^1, \\ \Delta w_k^2 = \tilde{a}^2 - \mathcal{T}_k[\Phi^\star(\tilde{a}^1)]. \end{cases}$$

It remains to provide an efficient manner to compute $\mathcal{T}_k[\cdot]$. Since $\mathcal{T}_k$ is defined as the operator form of $T = \tilde{P}_k L_k \tilde{P}_k^\top$, we have

$$\mathcal{T}_k[S] = P_k(\Psi_k \circ (P_k^\top S P_k))P_k^\top,$$

where $\Psi_k$ is determined by $\mu_k$ and $\Omega_k$. Indeed, we have

$$\Omega_k = \begin{pmatrix} E_{\alpha_k \alpha_k} & \eta_{\alpha_k \bar{\alpha}_k} \\ \eta_{\alpha_k \bar{\alpha}_k}^\top & 0 \end{pmatrix} \implies \Psi_k = \begin{pmatrix} \frac{1}{\mu_k}E_{\alpha_k \alpha_k} & \xi_{\alpha_k \bar{\alpha}_k} \\ \xi_{\alpha_k \bar{\alpha}_k}^\top & 0 \end{pmatrix},$$

where $\xi_{ij} = \frac{\eta_{ij}}{\mu_k+1-\eta_{ij}}$ for all $(i,j) \in \alpha_k \times \bar{\alpha}_k$. Following Zhao et al. [68], we use the decomposition $\mathcal{T}_k[S] = G + G^\top$ where $U = P_k(:, \alpha_k)^\top S$ and

$$G = P_k(:, \alpha_k)(\frac{1}{2\mu_k}(UP_k(:, \alpha_k))P_k(:, \alpha_k)^\top + \xi_{\alpha_k \bar{\alpha}_k} \circ (UP_k(:, \bar{\alpha}_k))P_k(:, \bar{\alpha}_k)^\top).$$

The number of flops required to compute $\mathcal{T}_k[S]$ is $8|\alpha_k|n^2$. For the case of $|\alpha_k| > \bar{\alpha}_k$, we compute $\mathcal{T}_k[S]$ via $\mathcal{T}_k[S] = \frac{1}{\mu_k}S - P_k((\frac{1}{\mu_k}E - \Psi_k) \circ (P_k^\top S P_k))P_k^\top$ using $8|\bar{\alpha}_k|n^2$ flops. This demonstrates that we can obtain an approximate solution of Eq. (3.2) efficiently whenever $|\alpha_k|$ or $|\bar{\alpha}_k|$ is small. We present the scheme for computing an approximate Newton direction in Algorithm 1.

**Adaptive strategy.** We propose a rule for updating $\theta_k$ where $\mu_k = \theta_k \|r_k\|$ is defined in Eq. (3.2). Indeed, we compute $\rho_k = -\langle R(w_k), \Delta w_k \rangle$ and use it to update $\theta_{k+1}$. The update rule is summarized as follows:

$$\theta_{k+1} = \begin{cases} \max\{\underline{\theta}, \beta_0 \theta_k\}, & \text{if } \rho_k \geq \alpha_2\|\Delta w_k\|^2, \\ \beta_1 \theta_k, & \text{if } \alpha_1\|\Delta w_k\|^2 \leq \rho_k < \alpha_2\|\Delta w_k\|^2, \\ \min\{\overline{\theta}, \beta_2 \theta_k\}, & \text{otherwise.} \end{cases} \tag{3.4}$$

where $\beta_0 < 1, \beta_1, \beta_2 > 1$ and $\underline{\theta}, \overline{\theta} > 0$.

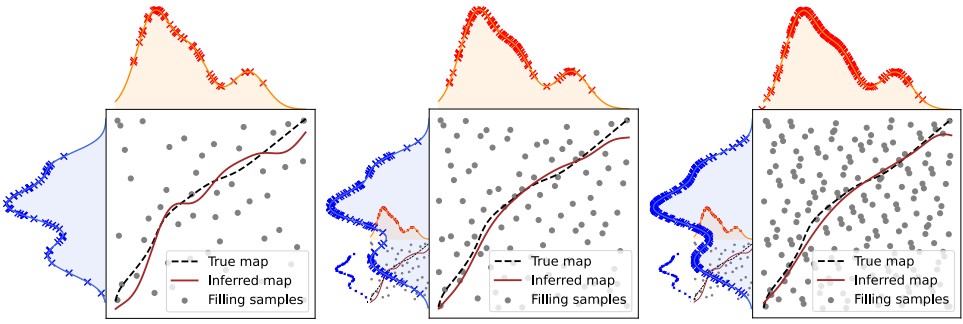

Figure 1: Visualization of the OT map with $n_{\text{sample}} = n \in \{50, 100, 200\}$.

**Main scheme.** We summarize the complete scheme of our new algorithm in Algorithm 2. Indeed, we generate a sequence of iterates by alternating between extragradient (EG) method [17, 6] and the aforementioned regularized SSN method.

Note that we maintain one auxiliary sequence of iterates $\{v_k\}_{k \geq 0}$. This sequence is directly generated by the EG method for solving the min-max optimization problem in Eq. (2.5) and is used to safeguard the regularized SSN method to achieve a global convergence rate. More specifically, we start with $v_0 = w_0 \in \mathbb{R}^n \times \mathcal{S}_+^n$ and perform the $k^{\text{th}}$ iteration as follows,

1. Update $v_{k+1}$ from $v_k$ using one-step EG.

2. Update $\tilde{w}_{k+1}$ from $w_k$ using one-step regularized SSN.

3. Set $w_{k+1} = \tilde{w}_{k+1}$ if $\|R(\tilde{w}_{k+1})\| \leq \|R(v_{k+1})\|$ and $w_{k+1} = v_{k+1}$ otherwise.

In our experiment, we find that the main iterates are mostly generated by regularized SSN steps and the whole algorithm converges at a superlinear rate. This phenomenon is quite intuitive: if the initial point is sufficiently close to one nondegenerate optimal solution, the regularized SSN method can achieve the similar quadratic convergence rate (cf. Theorem 3.4) as shared by other SSN methods in the existing literature [35, 18, 1]. The detailed analysis will be provided in the appendix.

**Main results.** We establish the convergence guarantee of Algorithm 2 in the following theorems.

**Theorem 3.3** *Suppose that $\{w_k\}_{k \geq 0}$ is a sequence of iterates generated by Algorithm 2. Then, the residuals of $\{w_k\}_{k \geq 0}$ converge to 0 at a rate of $1/\sqrt{k}$, i.e., $\|R(w_k)\| = O(1/\sqrt{k})$.*

**Theorem 3.4** *Suppose that $\{w_k\}_{k \geq 0}$ is a sequence of iterates generated by Algorithm 2. Then, the residuals of $\{w_k\}_{k \geq 0}$ converge to $\bar{0}$ at a quadratic rate if the initial point $w_0$ is sufficiently close to $w^\star$ with $R(w^\star) = \bar{0}$ and every element of $\partial R(w^\star)$ is invertible.*

**Remark 3.5** *In the context of constrained convex-concave min-max optimization problem, Cai et al. [6] proved the $O(1/\sqrt{k})$ last-iterate convergence rate of the EG, matching the lower bounds [24, 23]. Since the kernel-based OT estimation can be solved as a min-max problem, the global convergence rate in Theorem 3.3 demonstrates the efficiency of Algorithm 2. It remains unclear whether or not we can improve the convergence result by exploring special structure of Eq. (2.5).*

# 4 Experiments

We present the results of experiments that evaluate the kernel-based OT estimation with our algorithm. The baseline approach is the short-step interior-point method [59]; we exclude the gradient-based method [38] from our experiment since it only solves the relaxation model. All the experiments were conducted on a MacBook Pro with an Intel Core i9 2.4GHz and 16GB memory.

Following the setup in Vacher et al. [59], we draw $n_{\text{sample}}$ samples from $\mu$ and $n_{\text{sample}}$ samples from $\nu$, where $\mu$ is a mixture of 3 $d$-dimensional Gaussian distributions and $\nu$ is a mixture of 5 $d$-dimensional Gaussian distributions. Then, we sample $n$ filling samples from a $2d$ Sobol sequence. We also set the bandwidth $\sigma^2 = 0.01$ and parameters $\lambda_1 = \frac{1}{n}$ and $\lambda_2 = \frac{1}{\sqrt{n_{\text{sample}}}}$. Focusing on the case of $d = 1$ (i.e., 1-dimensional setting), we report the visualization results in Figure 1 and 2 and find that the inferred OT map will be closer the the true OT map as the number of filling points and data samples increase.

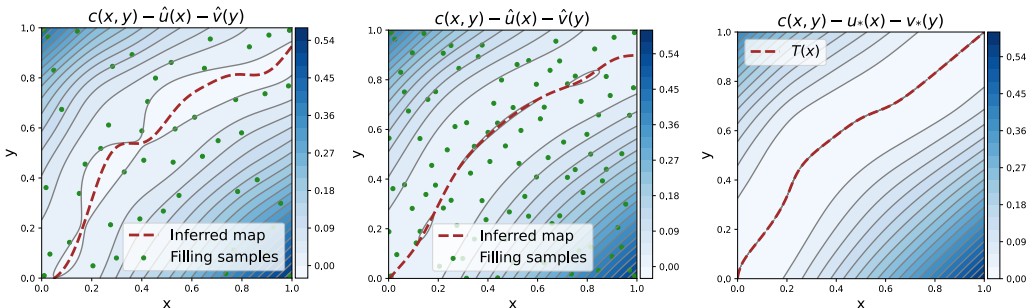

Figure 2: Visualization of the constraint with $n_{\text{sample}} = n \in \{50, 100\}$. The right one is ground truth.

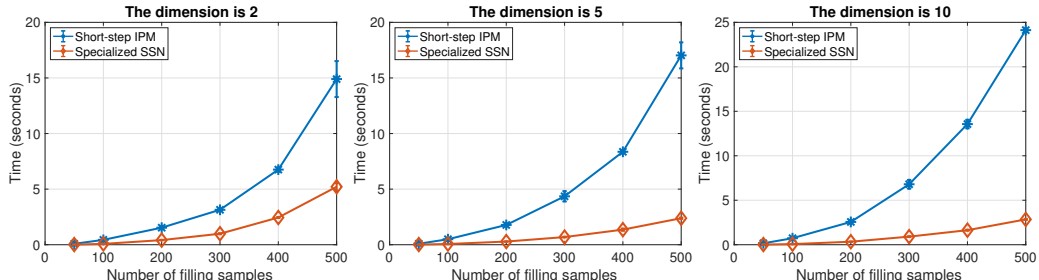

Figure 3: Comparisons of mean computation time of IPM and our algorithm on CPU time.

By varying the dimension $d \in \{2, 5, 10\}$, we also report the computation efficiency results in Figure 3. It indicates that the our new algorithm is more efficient than the IPM as the number of filling points increases, with smaller variance in computation time (seconds).

The experiments comparing kernel-based OT estimators with plug-in OT estimators on synthetic datasets have been conducted before [59, 38] and the results demonstrate that the kernel-based OT estimators behave better when the number of samples is small. Here, we repeat such experiment but using the real-world 4i datasets from Bunne et al. [5], which contains single-cell perturbed responses, and which include the unperturbed cells and cells subject to drug perturbations. Our experiments are conducted on 15 datasets with different drug perturbations.

Due to space limit, we defer the results to Appendix G (see Figure 4). We can see that the kernel-based OT estimators computed by our algorithm achieve satisfactory performance and behave better in most cases when the number of training samples is small; in particular, they better on 6 datasets, comparable on 5 datasets and worse on 4 datasets. Note that OTT computes the entropic regularized plug-in OT estimators and is heavily optimized to effectively handle noisy data. Therefore, it would be no surprise that OTT outperforms our algorithm when the number of training samples is sufficient. However, the kernel-based OT estimation still provides a fairly effective alternative when the number of training samples is small, which is consistent with the previous observations on synthetic data [59, 38]. Our results also validate the effectiveness of our algorithm for computing kernel-based OT estimators.

## 5   Concluding Remarks

In this paper, we propose a nonsmooth equation model for computing kernel-based OT estimators and show that it has a special problem structure, allowing it to be solved in an efficient manner using semismooth Newton method. In particular, we propose a specialized semismooth Newton method that achieves low per-iteration computational cost by exploiting the special problem structure, and prove a global sublinear convergence rate and a local quadratic convergence rate under standard regularity conditions. Preliminary experimental results on synthetic datasets show that our algorithm is more efficient than the short-step interior-point method [59], and the results on real data demonstrate the effectiveness of our algorithm. Future work includes the applications of kernel-based OT estimators to deep generative models and other real-world problems.

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
