## A Further Related Works on Semismooth Newton Method

Semismooth Newton methods [58] are a modern class of remarkably powerful and versatile algorithms for solving constrained optimization problems with partial differential equations, variational inequalities, and related problems.

The notion of semi-smoothness was originally introduced by Mifflin [37] for real-valued functions and later extended to vector-valued mappings by Qi and Sun [47]. A pioneering work on the semismooth Newton method was due to Solodov and Svaiter [54], in which the authors proposed a globally convergent Newton method by exploiting the structure of monotonicity and established a local superlinear convergence rate under the conditions that the generalized Jacobian is semismooth and nonsingular at the global optimal solution. The convergence rate guarantee was later extended in Zhou and Toh [69] to the setting where the generalized Jacobian is not necessarily nonsingular.

Recently, the semismooth Newton method has received significant amount of attention due to its wide success in solving several structured convex problems to a high accuracy. In particular, such approach has been successfully applied to solving large-scale SDPs [68, 67], LASSO [30], nearest correlation matrix estimation [45], clustering [61], sparse inverse covariance selection [65] and composite convex minimization [64]. The closest works to ours is Liu et al. [33], who developed a fast semismooth Newton method to compute the plug-in optimal transport estimator by exploring the sparsity and multiscale structure of its linear programming formulation. To the best of our knowledge, this paper is the first to apply the semismooth Newton method to computing the kernel-based optimal transport estimator and prove the convergence rate guarantees.

## B Proof of Proposition 2.3

We first prove that $\hat{\gamma}$ is an optimal solution of Eq. (2.4) if $\hat{w} = (\hat{\gamma}, \hat{X})$ satisfies $R(\hat{w}) = 0$ for some $\hat{X} \succeq 0$. Indeed, by the definition of $R$ from Eq. (2.6), we have

$$\frac{1}{2\lambda_2} Q\hat{\gamma} - \frac{1}{2\lambda_2} z - \Phi(\hat{X}) = 0, \tag{B.1}$$

and

$$\hat{X} - \text{proj}_{\mathcal{S}_+^n}(\hat{X} - (\Phi^\star(\hat{\gamma}) + \lambda_1 I)) = 0. \tag{B.2}$$

By the definition of $\text{proj}_{\mathcal{S}_+^n}$, we have

$$\langle X - \text{proj}_{\mathcal{S}_+^n}(\hat{X} - (\Phi^\star(\hat{\gamma}) + \lambda_1 I)), \text{proj}_{\mathcal{S}_+^n}(\hat{X} - (\Phi^\star(\hat{\gamma}) + \lambda_1 I)) - \hat{X} + (\Phi^\star(\hat{\gamma}) + \lambda_1 I) \rangle \geq 0 \text{ for all } X \succeq 0.$$

Plugging Eq. (B.2) into the above inequality yields that

$$\langle X - \hat{X}, \Phi^\star(\hat{\gamma}) + \lambda_1 I \rangle \geq 0 \text{ for all } X \succeq 0.$$

By setting $X = 0$ and $X = 2\hat{X}$, we have $\langle \hat{X}, \Phi^\star(\hat{\gamma}) + \lambda_1 I \rangle \leq 0$ and $\langle \hat{X}, \Phi^\star(\hat{\gamma}) + \lambda_1 I \rangle \geq 0$. Thus, we have

$$\langle \hat{X}, \Phi^\star(\hat{\gamma}) + \lambda_1 I \rangle = 0, \quad \langle X, \Phi^\star(\hat{\gamma}) + \lambda_1 I \rangle \geq 0 \text{ for all } X \succeq 0. \tag{B.3}$$

Suppose that $\gamma \in \mathbb{R}^n$ satisfies that $\Phi^\star(\gamma) + \lambda_1 I \succeq 0$, we have

$$
\begin{aligned}
0 &\overset{\text{(B.1)}}{=} (\gamma - \hat{\gamma})^\top \left( \frac{1}{2\lambda_2} Q\hat{\gamma} - \frac{1}{2\lambda_2} z - \Phi(\hat{X}) \right) \\
&= \left( \frac{1}{4\lambda_2} \gamma^\top Q\gamma - \frac{1}{2\lambda_2} \gamma^\top z \right) - \left( \frac{1}{4\lambda_2} \hat{\gamma}^\top Q\hat{\gamma} - \frac{1}{2\lambda_2} \hat{\gamma}^\top z \right) - \frac{1}{4\lambda_2} (\gamma - \hat{\gamma})^\top Q(\gamma - \hat{\gamma}) - (\gamma - \hat{\gamma})^\top \Phi(\hat{X}) \\
&\leq \left( \frac{1}{4\lambda_2} \gamma^\top Q\gamma - \frac{1}{2\lambda_2} \gamma^\top z \right) - \left( \frac{1}{4\lambda_2} \hat{\gamma}^\top Q\hat{\gamma} - \frac{1}{2\lambda_2} \hat{\gamma}^\top z \right) - (\gamma - \hat{\gamma})^\top \Phi(\hat{X})
\end{aligned}
$$

Since $\Phi^\star$ is the adjoint of $\Phi$, we have $(\gamma - \hat{\gamma})^\top \Phi(\hat{X}) = \langle \hat{X}, \Phi^\star(\gamma) - \Phi^\star(\hat{\gamma}) \rangle$. By combining this equality with $\Phi^\star(\gamma) + \lambda_1 I \succeq 0$ and the first equality in Eq. (B.3), we have

$$(\gamma - \hat{\gamma})^\top \Phi(\hat{X}) = \langle \hat{X}, \Phi^\star(\gamma) + \lambda_1 I \rangle - \langle \hat{X}, \Phi^\star(\hat{\gamma}) + \lambda_1 I \rangle \geq 0.$$

Thus, we have

$$0 \leq \left( \frac{1}{4\lambda_2} \gamma^\top Q\gamma - \frac{1}{2\lambda_2} \gamma^\top z + \frac{q^2}{4\lambda_2} \right) - \left( \frac{1}{4\lambda_2} \hat{\gamma}^\top Q\hat{\gamma} - \frac{1}{2\lambda_2} \hat{\gamma}^\top z + \frac{q^2}{4\lambda_2} \right).$$

486  Combining the above inequality with the second inequality in Eq. (B.3) yields the desired result.

487  It suffices to prove that satisfies $R(\hat{w}) = 0$ for some $\hat{X} \succeq 0$ if $\hat{\gamma}$ is an optimal solution of Eq. (2.4).

488  Indeed, we follow Definition 2.1 and write that $\sum_{i=1}^{n} \hat{\gamma}_i \Phi_i \Phi_i^\top + \lambda_1 I \succeq 0$ and

$$\tfrac{1}{4\lambda_2} \hat{\gamma}^\top Q \hat{\gamma} - \tfrac{1}{2\lambda_2} \hat{\gamma}^\top z + \tfrac{q^2}{4\lambda_2} \leq \tfrac{1}{4\lambda_2} \gamma^\top Q \gamma - \tfrac{1}{2\lambda_2} \gamma^\top z + \tfrac{q^2}{4\lambda_2},$$

489  for all $\gamma \in \mathbb{R}^n$ satisfying that $\sum_{i=1}^{n} \gamma_i \Phi_i \Phi_i^\top + \lambda_1 I \succeq 0$. Then, the KKT condition guarantees that

490  there exists some $\hat{X} \succeq 0$ satisfying that

$$\begin{array}{rcl}
\sum_{i=1}^{n} \hat{\gamma}_i \Phi_i \Phi_i^\top + \lambda_1 I & \succeq & 0, \\
\tfrac{1}{2\lambda_2} Q \hat{\gamma} - \tfrac{1}{2\lambda_2} z - \Phi(\hat{X}) & = & 0, \\
\langle \hat{X}, \Phi^\star(\hat{\gamma}) + \lambda_1 I \rangle & = & 0.
\end{array} \tag{B.4}$$

491  The first and third inequalities guarantee that

$$\langle X - \hat{X}, \Phi^\star(\hat{\gamma}) + \lambda_1 I \rangle \geq 0 \text{ for all } X \succeq 0.$$

492  By letting $X = \text{proj}_{\mathcal{S}_+^n}(\hat{X} - (\Phi^\star(\hat{\gamma}) + \lambda_1 I))$, we have

$$\langle \text{proj}_{\mathcal{S}_+^n}(\hat{X} - (\Phi^\star(\hat{\gamma}) + \lambda_1 I)) - \hat{X}, \Phi^\star(\hat{\gamma}) + \lambda_1 I \rangle \geq 0. \tag{B.5}$$

493  Recall that the definition of $\text{proj}_{\mathcal{S}_+^n}$ implies that

$$\langle X - \text{proj}_{\mathcal{S}_+^n}(\hat{X} - (\Phi^\star(\hat{\gamma}) + \lambda_1 I)), \text{proj}_{\mathcal{S}_+^n}(\hat{X} - (\Phi^\star(\hat{\gamma}) + \lambda_1 I)) - \hat{X} + (\Phi^\star(\hat{\gamma}) + \lambda_1 I) \rangle \geq 0 \text{ for all } X \succeq 0.$$

494  By letting $X = \hat{X}$, we have

$$\|\text{proj}_{\mathcal{S}_+^n}(\hat{X} - (\Phi^\star(\hat{\gamma}) + \lambda_1 I)) - \hat{X}\|^2 \leq \langle \hat{X} - \text{proj}_{\mathcal{S}_+^n}(\hat{X} - (\Phi^\star(\hat{\gamma}) + \lambda_1 I)), \Phi^\star(\hat{\gamma}) + \lambda_1 I \rangle \overset{(B.5)}{\leq} 0.$$

495  Combining the above inequality with the second equality in Eq. (B.4) yields that

$$\tfrac{1}{2\lambda_2} Q \hat{\gamma} - \tfrac{1}{2\lambda_2} z - \Phi(\hat{X}) = 0, \qquad \hat{X} - \text{proj}_{\mathcal{S}_+^n}(\hat{X} - (\Phi^\star(\hat{\gamma}) + \lambda_1 I)) = 0.$$

496  Combining these inequalities with the definition of $R$ implies $R(\hat{w}) = 0$ and hence the desired result.

## C   Proof of Proposition 3.1

498  The strong semismoothness of $R$ follows from the derivation given in Sun and Sun [56] to establish
499  the semismoothness of projection operators. Indeed, the projection over a positive semidefinite cone
500  is guaranteed to be strongly semismooth [56, Corollary 4.15]. Thus, we have that $\text{proj}_{\mathcal{S}_+^n}(\cdot)$ is strongly
501  semismooth. Since the strong semismoothness is closed under scalar multiplication, summation and
502  composition, the residual map $R$ is strongly semismooth.

## D   Proof of Lemma 3.2

504  As stated in Lemma 3.2, we compute $Z_k = X_k - (\Phi^\star(\gamma_k) + \lambda_1 I)$ and the spectral decomposition of
505  $Z_k$ (cf. Eq. (3.1)) to obtain $P_k$, $\Sigma_k$ and the sets of the indices of positive and nonpositive eigenvalues
506  $\alpha_k$ and $\bar{\alpha}_k$. We then compute $\Omega_k$ using $\Sigma_k$, $\alpha_k$ and $\bar{\alpha}_k$ and finally obtain that $\tilde{P}_k = P_k \otimes P_k$ and
507  $\Gamma_k = \text{diag}(\text{vec}(\Omega_k))$. Thus, we can write the matrix form of $\mathcal{J}_k + \mu_k I$ as

$$J_k + \mu_k I = \begin{pmatrix} \tfrac{1}{2\lambda_2} Q + \mu_k I & -A \\ \tilde{P}_k \Gamma_k \tilde{P}_k^\top A^\top & \tilde{P}_k((\mu_k + 1)I - \Gamma_k)\tilde{P}_k^\top \end{pmatrix}.$$

508  For simplicity, we let $W_k = \tilde{P}_k \Gamma_k \tilde{P}_k^\top$ and $D_k = \tilde{P}_k((\mu_k + 1)I - \Gamma_k)\tilde{P}_k^\top$. Then, the Schur
509  complement trick implies that

$$\begin{aligned}
(J_k + \mu_k I)^{-1} &= \begin{pmatrix} \tfrac{1}{2\lambda_2} Q + \mu_k I & -A \\ W_k A^\top & D_k \end{pmatrix}^{-1} \\
&= \begin{pmatrix} I & 0 \\ -D_k^{-1} W_k A^\top & I \end{pmatrix} \begin{pmatrix} (\tfrac{1}{2\lambda_2} Q + \mu_k I + A D_k^{-1} W_k A^\top)^{-1} & 0 \\ 0 & D_k^{-1} \end{pmatrix} \begin{pmatrix} I & A D_k^{-1} \\ 0 & I \end{pmatrix}.
\end{aligned}$$

510 Define $T_k = \tilde{P}_k L_k \tilde{P}_k^\top$ where $L_k$ is a diagonal matrix with $(L_k)_{ii} = \frac{(\Gamma_k)_{ii}}{\mu_k + 1 - (\Gamma_k)_{ii}}$ and $(\Gamma_k)_{ii} \in (0, 1]$

511 is the $i^{\text{th}}$ diagonal entry of $\Gamma_k$. By the definition of $W_k$ and $D_k$, we have $D_k^{-1} = \frac{1}{\mu_k + 1}(I + T_k)$ and

512 $D_k^{-1} W = T_k$. Using these two identities, we can further obtain that

$$(J_k + \mu_k I)^{-1}$$
$$= \begin{pmatrix} I & 0 \\ -T_k A^\top & I \end{pmatrix} \begin{pmatrix} (\frac{1}{2\lambda_2} Q + \mu_k I + A T_k A^\top)^{-1} & 0 \\ 0 & \frac{1}{\mu_k + 1}(I + T_k) \end{pmatrix} \begin{pmatrix} I & \frac{1}{\mu_k + 1}(A + A T_k) \\ 0 & I \end{pmatrix}.$$

513 This completes the proof.

## E   Proof of Theorem 3.3

515 We can see from the scheme of Algorithm 2 that

$$\|R(w_k)\| \leq \|R(v_k)\| \quad \text{for all } k \geq 0,$$

516 where the iterates $\{v_k\}_{k \geq 0}$ are generated by applying the extragradient (EG) method for solving the

517 min-max optimization problem in Eq. (2.5). We also have that Cai et al. [6, Theorem 3] guarantees

518 that $\|R(v_k)\| = O(1/\sqrt{k})$. Putting these pieces together yields that

$$\|R(w_k)\| = O(1/\sqrt{k}).$$

519 This completes the proof.

## F   Proof of Theorem 3.4

521 We analyze the convergence property for one-step SSN step as follows,

$$w_{k+1} = w_k + \Delta w_k,$$

522 where $\mu_k = \theta_k \|R(w_k)\|$ and

$$\|(\mathcal{J}_k + \mu_k \mathcal{I})[\Delta w_k] + R(w_k)\| \leq \tau \min\{1, \kappa \|R(w_k)\|\|\Delta w_k\|\}. \tag{F.1}$$

523 Since $R$ is strongly smooth (cf. Proposition 3.1), we have

$$\frac{\|R(w + \Delta w) - R(w) - \mathcal{J}[\Delta w]\|}{\|\Delta w\|^2} \leq C, \quad \text{as } \Delta w \to 0.$$

524 Since $w_0$ is sufficiently close to $w^\star$ with $R(w^\star) = 0$ and the global convergence guarantee holds (cf.

525 Theorem 3.3), we have

$$\|R(w_k + \Delta w_k) - R(w_k) - \mathcal{J}_k[\Delta w_k]\| \leq 2C\|\Delta w_k\|^2.$$

526 which implies that

$$\|R(w_{k+1})\| = \|R(w_k + \Delta w_k)\| \leq \|R(w_k) + \mathcal{J}_k[\Delta w_k]\| + 2C\|\Delta w_k\|^2. \tag{F.2}$$

527 Plugging Eq. (F.1) into Eq. (F.2) yields that

$$\begin{aligned} \|R(w_{k+1})\| &\leq 2C\|\Delta w_k\|^2 + \mu_k \|\Delta w_k\| + \tau \kappa \|R(w_k)\|\|\Delta w_k\| \\ &\leq 2C\|\Delta w_k\|^2 + (\theta_k + \tau\kappa)\|R(w_k)\|\|\Delta w_k\|. \end{aligned} \tag{F.3}$$

528 Since $w_0$ is sufficiently close to $w^\star$ with $R(w^\star) = 0$ and every element of $\partial R(w^\star)$ is invertible, we

529 have that there exists some $\delta > 0$ such that

$$\|(\mathcal{J}_k + \mu_k \mathcal{I})[\Delta w_k]\| \geq \delta \|\Delta w_k\|.$$

530 The above equation together with Eq. (F.1) yields that

$$\|\Delta w_k\| \leq \frac{1}{\delta}\|(\mathcal{J}_k + \mu_k \mathcal{I})[\Delta w_k]\| \leq \frac{1}{\delta}(1 + \tau\kappa\|\Delta w_k\|)\|R(w_k)\|. \tag{F.4}$$

531 Plugging Eq. (F.4) into Eq. (F.3) yields that

$$\|R(w_{k+1})\| \leq \|R(w_k)\|^2 \left(\frac{2C}{\delta^2}(1 + \tau\kappa\|\Delta w_k\|)^2 + \frac{\theta_k + \tau\kappa}{\delta}(1 + \tau\kappa\|\Delta w_k\|)\right)$$

532 Note that $\|\Delta w_k\| \to 0$ and $\theta_k$ is bounded. Thus, we have $\|R(w_{k+1})\| = O(\|R(w_k)\|^2)$.

533 From the above arguments, we see that the quadratic convergence rate can be achieved if Algorithm 2

534 performs the SSN step when the initial iterate $x_0$ is sufficiently close to $w^\star$ with $R(w^\star) = 0$. This

535 implies that the safeguarding steps will never affect in local sense where Algorithm 2 generates

536 $\{w_k\}_{k \geq 0}$ by performing the SSN steps only. So Algorithm 2 achieves the local quadratic convergence.

537 This completes the proof.

## G    Additional Experimental Results

We describe our setup for the experiment on the real-world 4i datasets from Bunne et al. [5]. Indeed, we draw the unperturbed/perturbed samples for training from 15 cell datasets as follows,

$$x_1, \ldots, x_{n_{\text{sample}}} \sim \mu_{\text{unperturb}}, \quad y_1, \ldots, y_{n_{\text{sample}}} \sim \nu^k_{\text{perturb}} \text{ for } 1 \leq k \leq 15.$$

where $x_i, y_i \in \mathbb{R}^{48}$ and $\mu_{\text{unperturb}}, \nu^k_{\text{perturb}}$ represent the unperturbed cells and $k^{\text{th}}$ perturbed cells. For our algorithm, we generate 256 filling points and compare our method with the default implementation in OTT package [13]. Both our algorithm and OTT capture the OT map $T$ from training samples. Then, we fix the number of test samples as $m = 200$ and use the OT distance to measure the differences between $\frac{1}{m} \sum_{j=1}^{m} \delta_{T(\hat{x}_j)}$ and $\frac{1}{m} \sum_{j=1}^{m} \delta_{\hat{y}_j}$, where $\hat{x}_1, \ldots, \hat{x}_m \sim \mu_{\text{unperturb}}$ and $\hat{y}_1, \ldots, \hat{y}_m \sim \nu^k_{\text{perturb}}$ are unperturbed/perturbed samples for testing. Figure 4 reports the results on 15 single-cell datasets.

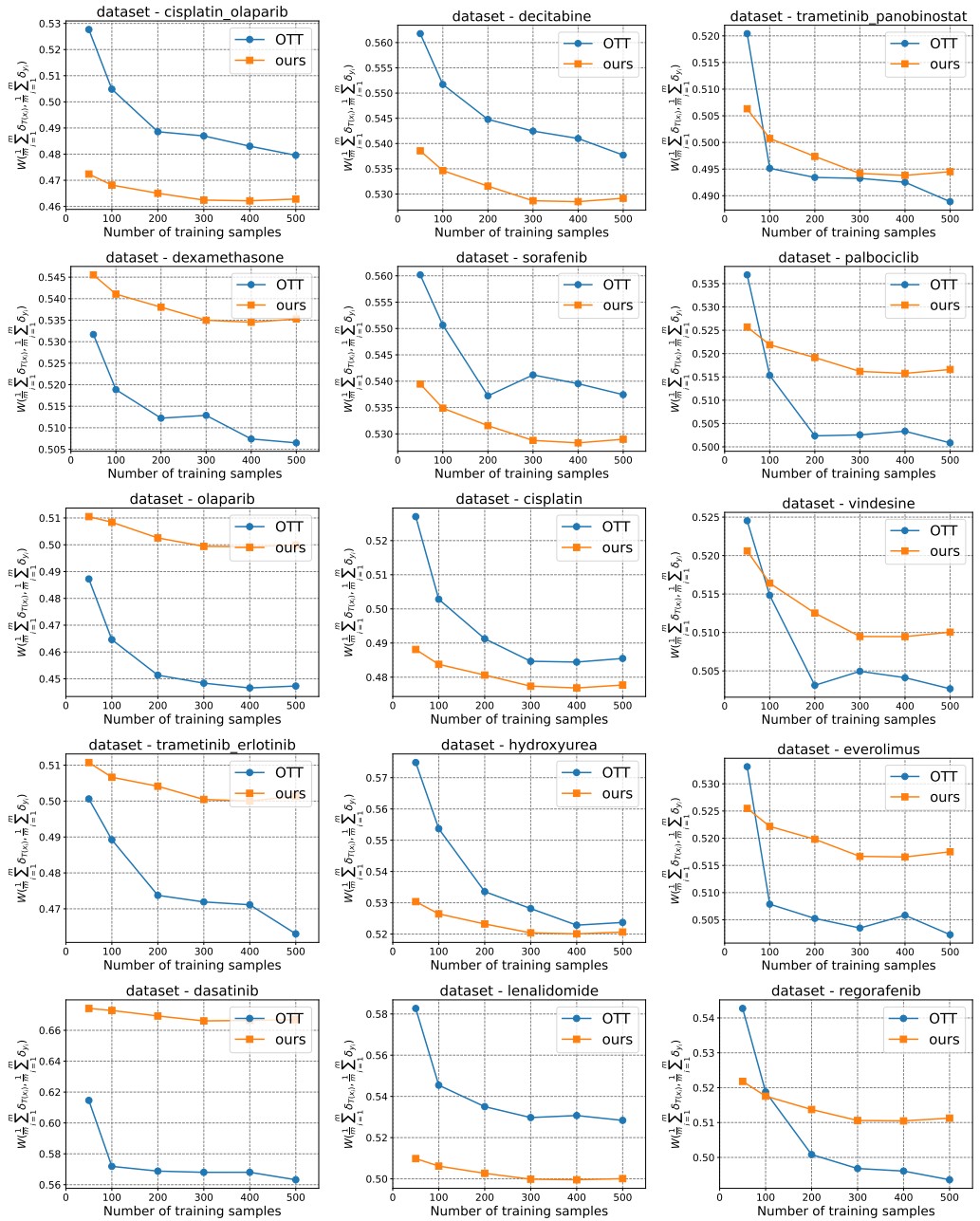

Figure 4: Performance of OTT and kernel-based OT estimators computed by our algorithm on 15 drug perturbation datasets. $X$-axis represent the number of training samples and $Y$-axis represents the error induced by OT map $T$ on test samples in terms of OT distance.