# OpenReview forum: "A Specialized Semismooth Newton Method for Kernel-Based Optimal Transport"
_NeurIPS.cc/2023/Conference — Submitted to NeurIPS 2023_

### Official Review · Reviewer_jxBx · 2023-06-12

**Soundness:** 3 good
**Presentation:** 2 fair
**Contribution:** 2 fair
**Rating:** 3
**Confidence:** 3

**Summary:**

The authors propose an implementation of Vacher et. al. (2021) based on a Semi-Smooth Newton (SSN) scheme. They reformulate their optimization problem as a root finding problem (Proposition 3.1) to which they apply the SSN scheme. They provide convergence guarantees (Theorem 3.3) that gives a $O(1/\sqrt{T})$ convergence rate where $T$ is the number of iterations and provide an efficient way to reduce the cost per iterations (l.184 - l.224). Then they provide numerical experiments to validate that their method is faster than the one proposed in Vacher et. al. (2021).

**Strengths:**

Trying to get a scalable version of kernel based OT is a very legit topic as current implementations are slow and impossible to run on real data sets. Indeed, recall that using an Interior-Point-Method, kernel based OT was solved with a precision $\epsilon$ in $O(n^{3.5} \log(n/\epsilon))$ time where $O(n^3)$ comes from the cost per iteration and $O(\sqrt{n}\log(n/\epsilon))$ is the number of iterations. In this paper, the main contribution is to get rid of the dependency of $n$ in the number of iterations which is indeed a desirable feature. From my understanding, the authors can solve kernel based OT with precision $\epsilon$ in $O(1/\epsilon^2)$ iterations.

**Weaknesses:**

I believe that the authors oversell the work. As can be deduced from my comment above, the proposed method requires $O(1/\epsilon^2)$ iterations for a precision $\epsilon$ while previous work requires $O(\sqrt{n}\log(n/\epsilon))$ iterations. When a high precision a sought after $\epsilon \to 0$, the proposed algorithm is indeed less efficient. The authors should have explicitly mentioned that.
Furthermore, nothing precise is said on the cost per-iteration which is a crucial component of the practical efficiency. We can vaguely guess that it is $O(n^3)$ but it is stated nowhere.

The overall writing is confusing, the whole part on the computational efficiency should be clearly stated in a theorem or a proposition.


**Questions:**

I am actually skeptical on the $O(1/\sqrt{T})$ convergence rate. What is the dependency in the regularizers $\lambda_1, \lambda_2$ and more generally in the condition number? In the case of Vacher et. al. (2021), there is little dependency in the condition number as they use an IPM-like method. Do SSN methods also weakly depend on the conditioning of the problem? Note that this is a crucial aspect as these regularizers implicitly depend on the number of samples $n$.
On the experimental side, it is claimed that the proposed method is faster. Yet which stopping criterion was used? Was it the same for both algorithms?

**Limitations:**

The authors do not compare with enough precision their algorithm with the existing one, both in theory and on practice.

---

> ### Author Rebuttal · Authors · 2023-08-06
>
> Thank you for your time and your input. We hope that with our answer below we will convince you about the merits of our work. Below, we reply to your main questions point-by-point and have included these discussions in the revised version of our paper.
>
> 1. **The proposed method requires $O(1/\epsilon^2)$ iterations while IPM requires iterations $O(\sqrt{n}\log(n/\epsilon))$. Considering the high precision, the proposed algorithm is indeed less efficient.**
>
>     We agree that our method is less efficient for the case of high precision (i.e., $\epsilon \rightarrow 0$) but argue that the better dependence on the sample size $n$ is more desirable since the large sample size is necessary to ensure better statistical approximation. In particular, the same authors of Vacher et. al. (2021) has mentioned in their subsequent paper (see [38], Page 11-12): *This method has two drawbacks: first, its cost is prohibitive when the number of samples becomes large, which is necessary to ensure better statistical approximation, and second, $\ldots$*
>
>     Thus, it is important to design the new kernel-based OT algorithms that have better dependence on $n$. Such discussion about the trade-off between $n$ and $1/\epsilon$ has occured in the community. Indeed, the plug-in OT estimation can be formulated as a linear optimization problem and solved by the specialized IPM within $O(\sqrt{n}\log(n/\epsilon))$ iterations. It can be also solved by Sinkhorn method within $O(1/\epsilon^2)$ iterations. Despite the worse dependence in terms of $1/\epsilon$, the Sinkhorn method has been recognized as more efficient than IPM in most cases since many OT application problems require low-accurate solution ($\epsilon \sim 10^{-2}$) when the sample size $n$ is very large. Along this direction, we provide a further step into the kernel-based OT algorithmic design and we hope that our idea may be useful more broadly.
>
>     We remark that our work does not downgrade the importance of IPM since our method becomes less efficient for small $\epsilon$. In our humble opinion, it seems promising to improve IPM by designing (1) the adaptive strategy; (2) the fast subproblem solvers as we have done for our method.
>
> 2. **Nothing precise is said on the per-iteration cost which is a crucial component of the practical efficiency. We can vaguely guess that it is $O(n^3)$ but it is stated nowhere.**
>
>     We agree that the per-iteration cost would be $O(n^3)$ at worst case but argue that it can be much cheaper in pratice. Indeed, the $O(n^3)$ cost comes from exactly solving the $n \times n$ linear system (see Page 7, Line 211). In Page 7, Line 212-214, we stated that this system can be *efficiently* solved by conjugate gradient (CG) method or symmetric quasi-minimal residual (QMR) method. In our experiment, we use CG to approximately solve this linear system and set the maximum iteration number as 20. Empirically, the average number of CG steps is less than 5. Also, the implementation of our method can be improved by exploring the structure Q, A and T_k, e.g., sparsity, but we have not incorporate it yet. In contrast, the linear system solving at each IPM step becomes severely ill-conditioned as the barrier parameter decreases and the matrix factorization has to be done exactly to achieve high precision. To summarize, our method suffers from the same per-iteration cost as IPM at worst case but can be more flexible and efficient from a practical viewpoint.
>
> 3. **What is the dependency in the regularizers $\lambda_1$, $\lambda_2$ and more generally in the condition number? Do SSN methods also weakly depend on the conditioning of the problem?**
>
>     The global rate of $O(1/\sqrt{T})$ is achieved since our method is at least as fast as the extragradient (EG) method for solving the min-max formulation (see Eq. (2.5)); indeed, our method alternates between EG and the regularized SSN method (see Line 232-238). We view Eq. (2.5) as a smooth and convex-concave min-max problem and know that the EG achieves the *optimal* last-iterate convergence rate of $O(1/\sqrt{T})$ (see Cai et al. [6, Theorem 3]). Such global rate depends on the smoothness parameter of Eq. (2.5) rather than the condition number of original formulation of Eq. (2.2). The explicit dependence on $\lambda_1$ and $\lambda_2$ is unknown since the results of Cai et al. [6] does not provide the dependence on all problem parameters. Nonetheless, our experiment has shown that our method behaves well when the sample size is medium (~500) which is sufficient for kernel-based OT estimation in most cases.
>
>     Similar to Newton methods which are key ingredients for IPM, the SSN methods enjoy the weak dependence on problem conditioning; see *A nonsmooth version of Newton’s method*. In Appendix F, the proof of Theorem 3.4 gives the dependence on problem parameters.
>
> 4. **On the experimental side, it is claimed that the proposed method is faster. Yet which stopping criterion was used? Was it the same for both algorithms?**
>
>      We apologize for the confusion we have created for not being specific. Indeed, we used the residue norm $\||R(w)\||$ (see Eq. (2.6)) as the measurement and terminated IPM and our method when $\||R(w)\||$ is below than the same threshold ($10^{-5}$). Notably, IPM can output the better solution than our method since the last iteration of IPM reduces $R(w)$ from $>10^{-5}$ to $\sim 10^{-7}$. However, the implementation from Vacher et. al. (2021) is based on short-step dual IPM and needs many iterations to reach $~10^{-4}$. In contrast, our method uses the adaptive strategy so needs much less iterations. Nonetheless, the IPM can be also improved using an adaptive strategy, e.g., Mehrotra's predict–correct rule, but this is beyond the scope of this paper.
>
> We thank you again for your detailed reading and your constructive input! We hope and trust that our replies have alleviated your concerns, and we look forward to an open-minded discussion if any such concerns remain.

---

> > ### Comment · Reviewer_jxBx · 2023-08-10
> > **Response**
> >
> > I thank the authors for taking the time. I will also respond point by point.
> >
> > 1. Sinkhorn is more popular than vanilla OT not because of the 1/eps**2 number of steps (which I believe is actually 1/eps) but because of the n^2 cost per step.
> >
> > 2. The system l.212 -214 is most likely ill conditioned, hence in theory, CG requires many steps to converge.
> >
> > 3. There is probably a dependence in the regularizers and it should have been clearly stated in the text.
> >
> > Overall I feel like that the contribution is more empirical than theoretical and I wished it had been clearly stated in the article.

---

> > > ### Author Response · Authors · 2023-08-11
> > > **Thanks for reacting to our rebuttal**
> > >
> > > Thanks for your prompt reply. We hope that with our answers below we can convince you further about the merits of our work.
> > >
> > > Please let us know if you have any other concerns, we will do our best to answer them.
> > >
> > > > **1. Sinkhorn is more popular than vanilla OT not because of the $1/\epsilon^2$ number of steps (which I believe is actually $1/\epsilon$) but because of the $n^2$ cost per step.**
> > >
> > > We do not think that your points are right. Indeed, "vanilla OT", i.e. the linear optimization algorithms (e.g. the network simplex method) can also achieve a per-iteration complexity of $n^2$ (see e.g. computational OT, Section 3.5.3, https://arxiv.org/abs/1803.00567). This is the effort required to check for a violating edge.
> > >
> > > To the best of our knowledge, the best bound on the iteration complexity is known as $1/\epsilon^2$ for Sinkhorn method and proved in the following paper: Computational optimal transport: Complexity by accelerated gradient descent is better than by Sinkhorn's algorithm in ICML 2018, https://proceedings.mlr.press/v80/dvurechensky18a.html. The improved bound of $1/\epsilon$ can be achieved by some other efficient methods, e.g., a gradient-based method (see the paper A direct $\tilde{O}(1/\epsilon)$ iteration parallel algorithm for optimal transport in NeurIPS 2019, https://proceedings.neurips.cc/paper_files/paper/2019/hash/024d2d699e6c1a82c9ba986386f4d824-Abstract.html) or a graph algorithm (see the paper A graph theoretic additive approximation of optimal transport in NeurIPS 2019, https://proceedings.neurips.cc/paper_files/paper/2019/hash/9b07f50145902e945a1cc629f729c213-Abstract.html). We would appreciate if you could provide the reference that proves the improved bound of $1/\epsilon$ for Sinkhorn method.
> > >
> > > > **2. The system l.212 -214 is most likely ill conditioned, hence in theory, CG requires many steps to converge.**
> > >
> > > We believe that the conditioning of linear systems will inevitably appear in Newton methods but this does not affect their value in both theory and practice. We have shown that our method was reliable in the experimental evaluation and CG works well (the preconditioning technique is used there). Compared to IPM, we find that our method based on semi-smooth Newton has better conditioning of linear systems in the experiment.
> > >
> > > > **3. There is probably a dependence in the regularizers and it should have been clearly stated in the text.**
> > >
> > > We would appreciate if you could clarify what you mean by "probably a dependence". Indeed, we have explained in the rebuttal why our method does not suffer from the small value of the regularizers. It would really help us if you could provide us with an example sentence that would clarify what you mean by "it should have been clearly stated in the text".
> > >
> > > > **4. Overall I feel like that the contribution is more empirical than theoretical and I wished it had been clearly stated in the article.**
> > >
> > > It is worth noting that our paper has indeed a practical purpose, hence it does blend both aspects. Our goal is efficiency, to open up new research directions exploiting this kernel-based OT approach, as seen e.g. in our abstract.
> > >
> > > *In this paper, we propose a nonsmooth equation model for kernel-based OT estimation and show that it can be efficiently solved via a specialized semismooth Newton (SSN) method. Indeed, by exploring the special problem structure, the per-iteration cost of performing one SSN step can be significantly reduced in practice. We also prove that our algorithm can achieve a global convergence rate of $O(1/\sqrt{k})$ and a local quadratic convergence rate under some standard regularity conditions. Finally, we demonstrate the effectiveness of our algorithm by conducing the experiments on both synthetic and real datasets.*

---

> > > > ### Comment · Reviewer_jxBx · 2023-08-11
> > > > **Response**
> > > >
> > > > 1. My bad yet when i refer to sec 5.3.2 the overall complexity is given by O(n^3 log(n)) so Sinkohrn does improve the computation by a linear factor even if, as you mentioned, the cost per iteration is the same.
> > > >
> > > > 3. You admitted in your first response that « the explicit dependence on the regularizers in unknown ». Maybe you observed in practice that small regularizers do not affect the convergence but you cannot guarantee it in theory. And I’m just saying that usually in optimization, there is no free lunch hence I suspect that a poor conditioning of the objective will result in a slower algorithm.
> > > >
> > > > 4. You say in your abstract that the cost per iteration can be significantly reduced so we expect a very detailed proof in the paper or paragraph in the paper but the crucial expensive step of l 212-214 is only mentioned super quickly. The details on CG are only given in the experiment section.

---

> > > > > ### Author Response · Authors · 2023-08-12
> > > > > **Thanks for reacting to our rebuttal again**
> > > > >
> > > > > Thanks for your prompt reply. We hope that with our answers below we can convince you further about the merits of our work.
> > > > >
> > > > > Please let us know if you have any other concerns, we will do our best to answer them.
> > > > >
> > > > > **1. My bad yet when I refer to sec 5.3.2 the overall complexity is given by $O(n^3 log(n))$ so Sinkohrn does improve the computation by a linear factor even if, as you mentioned, the cost per iteration is the same.**
> > > > >
> > > > > We would like to thank you for confirming that the network simplex method and Sinkhorn method achieve the same per-iteration complexity of $n^2$. Therefore, Sinkhorn method is more popular than vanilla OT because its iteration complexity is $O(\epsilon^{-2})$ that does improve the iteration complexity of network simplex method by a factor $n$ regardless of the worse dependence on $1/\epsilon$. The same logic can be used for comparing our method with the IPM. Indeed, the iteration complexity of our method is $O(\epsilon^{-2})$ that does improve the iteration complexity of IPM by a factor $\sqrt{n}$ regardless of worse dependence on $1/\epsilon$.
> > > > >
> > > > > **2. You admitted in your first response that « the explicit dependence on the regularizers is unknown ». Maybe you observed in practice that small regularizers do not affect the convergence but you cannot guarantee it in theory. And I’m just saying that usually in optimization, there is no free lunch hence I suspect that a poor conditioning of the objective will result in a slower algorithm.**
> > > > >
> > > > > Although we agree that there is no free lunch in optimization, we do not understand why you suspect that our method will suffer from a poor conditioning of the objective function. Indeed, we acknowledge that the explicit dependence on the regularizers is unknown but argue that our empirical results show that such dependence is weak. Moreover, similar to Newton methods which are key ingredients for IPM, the SSN methods enjoy the weak dependence on problem conditioning; see *A nonsmooth version of Newton’s method*. Therefore, we believe that our method can be an efficient candidate for solving kernel-based OT problems.
> > > > >
> > > > > **3. You say in your abstract that the cost per iteration can be significantly reduced so we expect a very detailed proof in the paper or paragraph in the paper but the crucial expensive step of l 212-214 is only mentioned super quickly. The details on CG are only given in the experiment section.**
> > > > >
> > > > > We apologize for the lack of clarity. Indeed, we say that the cost per iteration can be significantly reduced since we have shown that solving the linear system whose matrix form has size $(n^2+n) \times (n^2+n)$ (i.e., Eq. (3.2)) can be equivalently reduced to solving a much smaller linear system whose matrix form has size $n \times n$. Thus, the per-iteration computational cost would be significantly reduced. Such equivalent reduction is based on Lemma 3.2 whose proof is nontrivial and thus deferred to the appendix. This is one of our key contributions while the preconditioned CG used in our paper is quite standard and can be found in any textbook (see e.g., *Linear and Nonlinear Programming* by David Luenberger and Yinyu Ye).

---

> > > > > > ### Comment · Reviewer_jxBx · 2023-08-15
> > > > > > **Response**
> > > > > >
> > > > > > As long as you have not proven formally, or stated a precise theorem showing that your method does not suffer from the poor conditioning, I cannot consider that the complexity of your algorithm is indeed O(n^3/eps^2). In particular, I cannot consider that your work is an improvement over Vacher et al from a theoretical perspective.
> > > > > > Furthermore, I do not consider that the empirical contribution alone justifies the acceptance of your article.

---

> > > > > > > ### Author Response · Authors · 2023-08-16
> > > > > > > **Response and request for clarification**
> > > > > > >
> > > > > > > Thanks for your reply. We really appreciate your availability through this discussion phase and are very grateful for your time.
> > > > > > >
> > > > > > > Because we fail to understand your point, we kindly request a clarification.
> > > > > > >
> > > > > > > **1. As long as you have not proven formally, or stated a precise theorem showing that your method does not suffer from the poor conditioning, I cannot consider that the complexity of your algorithm is indeed $O(n^3/\epsilon^2)$. In particular, I cannot consider that your work is an improvement over Vacher et al from a theoretical perspective.**
> > > > > > >
> > > > > > > We humbly ask you to provide more details on what theorems, in your mind, would show that our method does not suffer from poor conditioning.
> > > > > > >
> > > > > > > The method in Vacher et al, 2021 is based on the short-step dual interior-point method.
> > > > > > >
> > > > > > > Their theorems (16 and 17) provide
> > > > > > > - a **statistical bound** for their estimator that is, indeed, conditioned on regularizers $\lambda_1$ and $\lambda_2$ (we also recover it, since we minimize the same problem),
> > > > > > > - a **computational bound** (the one you are requesting if we understand correctly) which **does not depend on conditioning and $\lambda_1$ and $\lambda_2$**. Their computational bound is $O(C+ E\ell + \ell^{3.5}\log (\ell/\tau)$ with $\ell$ the number of iterations (Theorems 16/17). That bound does not mention an explicit dependency on $\lambda_1$, $\lambda_2$ which are lumped inside constants.
> > > > > > >
> > > > > > > Yet, it seems that you still consider that their complexity bound is $O(n^{3.5}\log(n/\epsilon))$ and argue that it has no dependency on the condition number as they use an IPM-like method (please correct us if we misunderstand your points).
> > > > > > >
> > > > > > > Our method follows the very same approach, but is based on a semi-smooth Newton method instead, which also enjoys a weak dependence on problem conditioning; see *A nonsmooth version of Newton’s method*. Our empirical results also illustrate quite eloquently that, at the very least, such dependence is as weak in practice for our method as it is for Vacher et al. 2021.
> > > > > > >
> > > > > > > Therefore, we humbly request your clarification on comparing our method and the method in Vacher et al., 2021, because at this moment, we do not see what is missing in our analysis compared to that in Vacher et al 2021.
> > > > > > >
> > > > > > > **2. Furthermore, I do not consider that the empirical contribution alone justifies the acceptance of your article.**
> > > > > > >
> > > > > > > Although we agree that the exact dependency in the regularizers $\lambda_1$ and $\lambda_2$ is unknown, we do not understand why you consider that our contribution is only empirical.
> > > > > > >
> > > > > > > As we have stated in the abstract, we propose *a nonsmooth equation model* for kernel-based OT estimation, and show that it can be efficiently solved via *a specialized semismooth Newton (SSN) method*. We explore that problem's special structure, and show that the per-iteration cost of performing one SSN step can be significantly reduced in practice.
> > > > > > >
> > > > > > > Our methodology is new, and provides a new link between kernel-based OT and SSN methods. Taken together, this paper contributes to the broad landscape of computational OT by developing new algorithms with theoretical guarantee (i.e., global and local convergence rate) and practical implementation (i.e., empirical results).
> > > > > > >
> > > > > > > We are thankful for your time reading our response, and hope this clarifies a bit more the novelty of our paper.

---

### Official Review · Reviewer_7do7 · 2023-06-28

**Soundness:** 2 fair
**Presentation:** 2 fair
**Contribution:** 3 good
**Rating:** 6
**Confidence:** 3

**Summary:**

The authors focus on the problem of approximating OT numerically.
They focus on one approximated version of OT which leverages a Sum of Squares approximation to stratify both statistical guarantees and computational amenability.
While the first proposal to solve this SoS approximation relied on interior point methods, the authors focus on a semi-smooth Newton method.
It consists in considering KKT optimality as some equation $R(w)=0$ and solve this equation using Newton updates.
They derive the algorithm in this specific OT setting, and prove convergence guarantees and rates of their methods.
They show experiments on synthetic data to see the approximation impact, and compare with interior point methods.

**Strengths:**

This recent OT formulation satisfying statistical guarantees and being computationally amenable is an interesting quantity to estimate.
The proposal of the authors to propose another algorithm to estimate it and scale it to larger measures would increase the interest of this formulation to practitioners.


**Weaknesses:**

*The introduction is not precise enough*
- Line 25, the rate $O(n^{-1/2d})$ is actually worse than the original rate. I think the authors meant a rate $O(n^{-2/d})$.
- This is a secondary remark, but Line (31,32) another approach which attempts to regularize OT and ease computation is to consider mini batches of input data. I mention the work [FZFGC] and references therein if the authors wish to complement their introduction review.
- The citation [44] in your paper is irrelevant. It focuses on estimating the OT Monge map when it exists, which is not the problem of estimating the cost, which you consider. Also, saying ‘a specific […] estimator’ is a super vague formulation which should be made precise.
- Do the authors have references or precise rates to defend the assertion line 45-47 that « interior-point method is well known to be ineffective […] as the sample size increases » ? Similarly, do the authors have references that semi-smooth Newton method have better convergencce/scaling guarantees ?
- I do not understand the sentence « While there is an ongoing debate in the OT literature on the merits of computing the plug-in OT estimators v.s. kernel-based OT estimators […] ». Which debates it is ? On which aspect does it especially focus ? This sentence is too vague to be insightful.
- I do not understand the sentence Line 129 « kernel-based OT estimators are better when the sample size is small and the dimension is high ». Does that mean that the fewer samples we have, the better the approximation ?

*The semi-smooth Newton method is not clear to understand*
- Line 76, I think the authors should have introduced background knowledge on Semi-Smooth Newton methods instead of postponing them in the appendix. Furthermore, what is described by the authors is a review of previous contributions on this method, but no mathematical formulas are detailed. I would have put this part in the main body for related work, especially [33] which is exactly the same method as you, but for unregularized OT, and which you do not mention as related work. Lastly, to provide a self-contained and pedagogical description, I would have ideally wanted a brief description of SSN with a general framework, so that your work is an instantiation of this formulation.
- I think Definition 2.1 is not extremely useful as it is the definition of optimality in a minimization program, and you can remove it.
- Something that is not clear for me is whether some matrices are symmetric or not. First the set $S^n_+$ usually represent symmetric, positive matrice, but I see no symmetry in Line 152. The projection over $S^n_+$ of Equation (3.1) is true if Z is symmetric (or X in you context), but I see nowhere that X is assumed to be symmetric (or proved to be symmetric through iterations).  Line 192, you mention a Schur Complement trick to make the Jacobian symmetric, but when the matrix is asymmetric, there is no reason that the Schur complement is symmetric. All in all, the derivation of the method seems unclear and ill-posed. Could you please clarify on this ?
- Could you please define a quadratic rate of convergence using an equation ?

*Some experimental improvements to suggest*
You reproduce the experiments from [59], which is good to establish a comparison. However I think it could be made much clearer with some modifications.
- In Figure 2, I would be interested in seeing the point wise difference between $c - \hat{u} - \hat{v}$ and $c - u_*-v_*$. It would emphasize where the approximation is best performed using this estimator. Reproducing the same experiment using interior point method would be insightful.
- I don’t understand how time is estimated in Figure 3. Do you report the time to do a given number of iterations ? Is it the time to reach a given level of accuracy ? Without this I cannot make sure the comparison is fair.
- I think that reproducing Figure 6 from [59] would be insightful. My main question is that you focus on time and approximation error, but I would like to see the statistical error estimation as the number of samples grow. Reproducing this Figure (and comparing with interior point) would illustrate that your computational approach maintains the favorable statistical properties of this OT estimator.

[FZFGC] Fatras, K., Zine, Y., Flamary, R., Gribonval, R., & Courty, N. (2019). Learning with minibatch Wasserstein: asymptotic and gradient properties


**Questions:**

See my questions above. At the moment I advocate for rejection because I think the paper needs a significant amount of clarification w.r.t. their contributions, background and related work, such that I would not recommend publication in such state. However, I may have misunderstood parts of the paper, and I hope the authors will clarify this by answering my questions.


**Limitations:**

The authors adressed the societal impact of their work.

---

> ### Author Rebuttal · Authors · 2023-08-08
>
> Thank you for your time and your input. We hope that our answers below will convince you about the merits of our work. We answer your questions below one-by-one, and have included these discussions in a revised version of our paper.
>
> 1. **The rate $O(n^{-1/2d})$ should be a rate $O(n^{-2/d})$.** Fixed.
>
> 2. **Another approach that attempts to regularize OT [...] is to consider mini batches of input data, e.g., [FZFGC].** We have included the reference in our intro.
>
> 3. **[44] is irrelevant.** Because both Vacher [38] and our method yield dual potential **function** estimators, they can also produce OT map estimators using the Brenier formula (e.g. Eq. 44 in [38]), as used in Fig. 2. Hence, we believe [44] is a natural reference, but we will clarify.
>
> 4. **Any references or precise rates that defend line 45-47? Any references that semi-smooth Newton method has better convergence/scaling guarantees?** The reference that defends line 45-47 best is [38] from the same authors of [59]. They claimed in Page 11-12: *This method has two drawbacks: first, its cost is prohibitive when the number of samples becomes large, which is necessary to ensure better statistical approximation, and second, $\ldots$*. The drawback of short-step IPM was mentioned in *Interior-point methods* by Potra and Wright. The semi-smooth Newton method has better scaling guarantee for solving many problems [30,33,45,61,64,65,67,68], where [33] showed its power of solving large plug-in OT problem.
>
> 5. **I do not understand *While there is an ongoing debate in the OT literature on the merits of computing the plug-in OT estimators v.s. kernel-based OT estimators* Which debates it is ? On which aspect does it especially focus?** We will clarify this sentence. Plug-in estimators (e.g. LP based, Sinkhorn, or mini-batch) focus on the W distance. Kernel-based OT estimators estimate sufficiently smooth dual-potential **functions**. The "ongoing" debate refers to whether, to estimate the W distance, it might be better to "only" compute the objective of a (regularized) discrete problem, or to use samples to estimate dual (continuous) **functions**, and evaluate them on data. Plug-in OT estimators suffer from the curse of dimensionality, but are tractable for large $n$. In contrast, kernel-based OT estimators yield dimension free estimates, but solve a very costly conic optimization problem, which has only been approached using short-step IPM [38]. This motivates our more efficient method.
>
> 6. **I do not understand*kernel-based OT estimators are better [...] the fewer samples we have, the better the approximation?** We meant that kernel-based estimators are very efficient statistically speaking (dimension free rate), but are intractable for large sample sizes. Therefore, kernel-based OT estimators will be relevant when sample size $n$ is small (estimator is still tractable) and dimension $d$ is large (statistical rates are $O(n^{-2/d})$ and $O(n^{-1/2})$ for plug-in and kernel-based estimators, respectively).
>
> 7. **The authors should introduce background knowledge on SSN methods and put the review of previous contributions on this method in the main body, especially [33] which is exactly the same method as you, but for unregularized OT.** Due space constraints, we only gave a brief introduction to SSN methods for the broad NeurIPS audience, to focus in the main text in a clear presentation of our algorithm and results. Following your suggestion, we will include a general introduction to SSN in the appendix and highlight the differences between [33] and our work. If more space is allowed, we will move some of it back to the main text.
>
> 8. **Definition 2.1 is not extremely useful as it is the definition of optimality in a minimization program, and you can remove it.** Fixed.
>
> 9. **Something that is not clear for me is whether some matrices are symmetric or not.** We apologize for the lack of clarity. Indeed, $X$ is *symmetric* and positive semidefinite since it is defined as the dual variable for the constraint $\sum_{i=1}^n \gamma_i\Phi_i\Phi_i^\top + \lambda_1 I \succeq 0$; see Line 150. In the revised version, we rewrite $S_+^n = \\{X \in \mathbb{R}^{n \times n}: X^\top = X, X \succeq 0\\}$ and $X \in S_+^n$ instead of $X \succeq 0$ throughout our paper.
>
> 10. **Could you please define a quadratic rate of convergence using an equation?** We recall the residue norm $\||R(w)\||$ (see Eq. (2.6)) and define a quadratic rate as $\||R(w_{k+1})\|| \leq C\||R(w_k)\||^2$ for some constant $C > 0$.
>
> 11. **Figure 2: the pointwise difference between $c-\hat{u}-\hat{v}$ and $c-u_\star-v_\star$. It would emphasize where the approximation is best performed using this estimator.** This is a great idea, we will present this in the paper.
>
> 12. **Figure 3: the experimental setup for reporting time.** We used the residue norm $\||R(w)\||$ as the measurement and terminated IPM and our method when $\||R(w)\||$ is below than the same threshold ($10^{-4}$).
>
> 13. **Figure 6 from [59]: statistical error estimation rather than time and approximation error.** We agree that the statistical properties are worth investigating and reproduce Fig. 6 from [59] using our method and IPM. Both figures are almost indistinguishable, since both methods solve the same problem, and output sufficiently accurate solutions. We also would like to argue that the discovery of efficient computational methods often precedes other advances (applied or statistical). These are two distinct and complementary subjects. In our humble opinion, the contribution of our paper is computational, and studying computational aspects for kernel-based OT estimators with theoretical guarantee is necessary. We refer you to [38, 59] for the details on statistical properties.
>
> We thank you again for your detailed reading and your constructive input! We hope and trust that our replies have alleviated your concerns, and we look forward to an open-minded discussion if any such concerns remain.

---

> > ### Comment · Reviewer_7do7 · 2023-08-14
> > **Answer to Rebuttal**
> >
> > Dear Authors,
> >
> > I thank you for your rebuttal which clarified many points where I was thinking the formulation was too vague. In the revision you must clarify that you talk both about *statistical* and *computational* complexities, and that for now, the kernel or plug-in approach only enjoys a reasonable complexity on either one of these aspects. This would be much clearer for the sentences where I thought it was unclear.
> >
> > I would also insist on being self-contained on SSN methods, then instantiate the kernel-OT case inside a theorem. This would help understanding the principle of the method while not focusing on cumbersome notations which appear during the derivation of the OT setting. I suppose you have your reasons for processing in such manner, but it personnally complicated my understanding, given the ambiguity of variables which were either symmetric or not.
> >
> > Another question on which the authors can answer during this discussion is: Why can't we scale to more than 500 samples ? Compared to entropic OT which scales 'reasonnably' for 10^4 samples on GPUs, this seems very limiting, and you do not seem to solve this issue. Could you contextualize in the paper if it could be solved, or if it is due to solving a different approximation of OT problem ?
> >
> > Side remark which I noticed from checking your paper, but entropic OT does not suffer curse of dimensionality, but like kernel-OT, the constant might depend exponentially in the dimension, see e.g. [1]. In [44] and lines 41-42, it is just that the estimator is ill-posed, and is unable to exploit the regularity to break such curse of dimensionality to estimate a Monde map.
> >
> > For this reason I decide to increase my score. However, I would not personnaly advocate for a complete acceptance, because the requested modifications might need a new reviewing process due to their importance.
> >
> > [1] Genevay, A., Chizat, L., Bach, F., Cuturi, M., & Peyré, G. (2019, April). Sample complexity of sinkhorn divergences. In The 22nd international conference on artificial intelligence and statistics (pp. 1574-1583). PMLR.

---

> > > ### Author Response · Authors · 2023-08-15
> > > **Many thanks for answering our rebuttal before the deadline**
> > >
> > > We are very grateful for your timely response. Here are a few more answers:
> > >
> > > > **In the revision you must clarify that you talk both about statistical and computational complexities [...] This would be much clearer for the sentences where I thought it was unclear.**
> > >
> > > Yes, we heard you on this, and we will emphasize more strongly this trade-off, starting with the abstract, that we can change as
> > >
> > > *Recent works suggested that kernel-based OT [...] practice. In this paper, we propose [...]
> > >
> > > To
> > >
> > > *Recent works suggested that kernel-based OT estimators are more statistically efficient than plug-in OT estimators when comparing probability measures in high-dimensions [59]. However, this comes at a very steep computational price. These estimators are very costly, since their computation relies on the short-step interior-point method for which the required number of iterations is known to be large in practice. To improve on the scalability of these approaches, we propose in this paper a nonsmooth equation model for kernel-based OT estimation and show that it can be efficiently solved via a specialized semismooth Newton (SSN) method.*
> > >
> > > and more generally amend the introduction and background sections accordingly.
> > >
> > > > **I would also insist on being self-contained on SSN methods, then instantiate the kernel-OT case inside a theorem [...]**
> > >
> > > We agree you, and we will provide in Section 2.3 a self-contained introduction to SSN methods that will be about half a page.
> > >
> > >
> > > > **[...] Why can't we scale to more than 500 samples ? Compared to entropic OT which scales 'reasonably' for 10^4 samples on GPUs, this seems very limiting, and you do not seem to solve this issue. Could you contextualize in the paper if it could be solved, or if it is due to solving a different approximation of OT problem ?**
> > >
> > > This is indeed due to the fact that kernel-OT targets a completely different approximation of OT problems:
> > > - kernel-based OT solvers target a **functional** optimization problem, i.e. their solutions are directly dual potential functions that agree with prior smoothness assumptions (line 130).
> > > - by contrast, the Sinkhorn algorithm is a discrete solver that computes a transport **matrix**, or dual potential **vectors**. While its outputs have been recently used to recover dual potential functions (as in [Pooladian+21]), this is mostly an interpolation, a smooth c-transform of pointwise potential values, inspired by semi-discrete OT, and not the result of a functional optimization (as done with RKHS in kernel based OT).
> > >
> > > The approach by Vacher et. al was illustrated on a maximal number of 200 points. We propose experiments with 500 points. Our solver can scale reasonably (about 20 seconds) for 1000 points, but the IPM solver of Vacher does not, we will add this to the curves.
> > >
> > > While we agree this is still a bit small, it does start to open up some possibilities, using e.g. mini-batch OT. We will discuss this explicitly in the paper, and expand on Remark 2.2
> > >
> > > > **Side remark which I noticed from checking your paper, but entropic OT does not suffer curse of dimensionality, but like kernel-OT, the constant might depend exponentially in the dimension, see e.g. [1]. [...]**
> > >
> > > A discussion on entropic OT vs. kernel-OT is provided in the **State of the art** section of Vacher et al [59].
> > >
> > > We can add our summary of this: While the entropic OT rate you mention does indeed gives a $1/\sqrt{n}$ statistical dependency, this is only valid for a **fixed** regularization $\varepsilon>0$ level (i.e. statistical complexity assumes **regularized OT** between densities is the target ground truth). However, because that constant is dimensionality dependent, it will blow up exponentially fast to infinity as $\varepsilon\rightarrow 0$, if one wants to approximate the **non-regularized OT**. In that sense, entropic OT does not provide a dimension-free (event w.r.t. constants) way to compute non-regularized OT, and, more qualitatively, entropic OT can only make sense statistically for high regularizations (constant degrades exponentially fast). As $\varepsilon \rightarrow \infty$, one recovers the MMD complexity.
> > >
> > > By contrast, Vacher et al's show that kernel OT does not suffer from such a blow-up. While their constants do depend exponentially in $d$, they are **fixed**, and the rate in $1/\sqrt{n}$ to target non-regularized OT is valid.
> > >
> > > > **For this reason I decide to increase my score. However, I would not personnaly advocate for a complete acceptance, because the requested modifications might need a new reviewing process due to their importance.**
> > >
> > > We are very grateful for your score increase. We believe that the modifications you have requested only target the background section, only with clarifications (Vacher’s methods computational/statistical tradeoff + brief background on SSN). We do not need to add new material or original contributions to satisfy your requests. We humbly request your trust on carrying out these modifications.

---

> > > > ### Comment · Reviewer_7do7 · 2023-08-15
> > > > **Response**
> > > >
> > > > I thank you for these additionnal details. I decide to trust you on the amount of modifications, and to slightly increase my score again. However, I urge you to revise and clarify everything that was mentioned in the interest of your work's impact which is undermined by the vagueness of some parts.

---

> > > > > ### Author Response · Authors · 2023-08-15
> > > > > **We are very grateful for your time reading our last message.**
> > > > >
> > > > > Your concerns have been heard, we agree with them, and we will use them to improve the presentation of our draft. We will actively clarify the points you have raised in the introduction and background sections (notably statistical/computational tradeoff at stake in Vacher's method, and why it is important to improve on the latter). We will provide examples with 1k points.
> > > > >
> > > > > We thank you for kindly raising again your grade following this discussion.

---

### Official Review · Reviewer_bQMP · 2023-07-07

**Soundness:** 3 good
**Presentation:** 3 good
**Contribution:** 3 good
**Rating:** 7
**Confidence:** 4

**Summary:**

This paper focuses on investigating kernel-based optimal transport estimation. The approach involves reformulating the problem as a nonsmooth equation model and utilizing the semismooth Newton method to solve it. The study demonstrates that the associated residual mapping exhibits **strong semismooth** properties, ensuring the applicability of the semismooth Newton method. Additionally, it is verified that the subproblem within the semismooth Newton method is well-defined, as it is equivalent to solving an invertible symmetric linear system. Finally, the proposed algorithm is supported by both theoretical guarantees, including global and local rates, and numerical experiments that highlight its superiority.

**Strengths:**

1. The algorithm is highly practical and can be easily implemented. The paper provides clear instructions on solving the subproblem and updating the parameters, making it accessible for real-world applications.
2. The theoretical investigation is rigorous and well-founded. The authors define a suitable residual function and present both global and local convergence rates of the proposed semismooth Newton algorithm.
3. The numerical experiments provide compelling evidence of the algorithm's efficiency compared to existing methods. The results showcase the superior performance and computational advantages of the proposed approach, reinforcing its practical relevance and effectiveness.

**Weaknesses:**

1. The global convergence rate of the proposed algorithm is dependent on an auxiliary sequence of iterates, which adds extra computational complexity to the algorithm. It would be helpful to provide further clarification in line 238 regarding whether the condition $$w_{k+1}=v_{k+1}$$ always holds. If so, the proposed algorithm will reduced to extragradient method.

2. To show the power of semismooth Newton steps, the proposed algorithm should be compared with the pure extragradient method in numerical experiments.



**Questions:**

1. How to choose the hyperparameters $\alpha_1,\alpha_2$, and $\kappa$ etc. in Algorithm 2?
2. Is there any intuition to use the adaptive strategy (3.4)?
3. In the proof of Theorem 3.4, the auxiliary sequence $\{v_k\}$ is not considered. It seems that the strategy in line 238 cannot be neglected and the case $w_{k+1}=v_{k+1}$ needs to be precluded under the conditions of Theorem 3.4.


**Limitations:**

See weakness and questions for further details.

---

> ### Author Rebuttal · Authors · 2023-08-07
>
> Thank you for your encouraging comments and positive evaluation! We reply to your main questions point-by-point below and have included these discussions in the revised version of our paper.
>
> 1. **The global convergence rate of the proposed algorithm is dependent on an auxiliary sequence of iterates, which adds extra computational complexity to the algorithm. It would be helpful to provide further clarification in line 238 regarding whether the condition $w_{k+1} = v_{k+1}$ always holds. If so, the proposed algorithm will reduced to extragradient method.**
>
>     We agree that computing the auxiliary sequence results in extra cost but argue that such cost is less than that of performing one-step regularized SSN. In our experiment, we also find that the main iterates are mostly generated by regularized SSN steps and the whole algorithm converges at a superlinear rate (see Page 8, Line 239-240). Thus, we can compute such auxiliary sequence at the initial stage and then only perform the regularized SSN steps.
>
>     We claim that $w_{k+1} = v_{k+1}$ will not always holds. In Page 8, Line 240-243, we stated that, if the initial point is sufficiently close to one nondegenerate optimal solution, the regularized SSN method can achieve the quadratic convergence rate as shared by other SSN methods in the existing literature [35, 18, 1] (see also Theorem 3.4). In other words, if the current iterate $w_k$ is sufficiently close to one nondegenerate optimal solution, the regularized SSN step achieves a quadratic rate (like the second-order method, e.g., Newton method) while the EG step only achieves a linear rate (the EG method is the first-order method). This implies that the regularized SSN step can reduce the residue norm more than the EG step and $w_{k+1} = v_{k+1}$ will not hold. Since Theorem 3.4 guarantees the existence of such local region where the regularized SSN step outperforms the EG step, it suffices to stop computing the auxiliary sequence after the iterates enter the local region and perform the regularized SSN steps. This supports the use of early stopping strategy as mentioned above. However, we remark that the implementation is nontrivial since it is difficult to check whether or not the iterates enter the local region in practice. If we stop computing such auxiliary sequence too early, our method is likely to diverge.
>
> 2. **To show the power of semismooth Newton steps, the proposed algorithm should be compared with the pure extragradient method in numerical experiments.**
>
> 	We agree that it would be better to compare our method with the pure EG method but hope to mention that the power of regularized SSN steps has been partially shown in our experiment. In Page 8, Line 239-240, we stated that, in our experiment, we find that the main iterates are mostly generated by regularized SSN steps and the whole algorithm converges at a superlinear rate. Following your suggestion, we conduct the experiment and the preliminary results show that the pure EG method outperform our method at the initial stage due to its relatively cheaper per-iteration cost but only output a low-accurate solution compared to our method (see the attached pdf file).
>
> 3. **How to choose the hyperparameters in Algorithm 2?**
>
>     We apologize for the confusion we have created for not being specific. Indeed, we choose $\alpha_1=10^{-6}$, $\alpha_2 = 1.0$, $\beta_0 = 0.5$, $\beta_1 = 1.2$ and $\beta_2 = 5$ in our experiment.
>
> 4. **Is there any intuition to use the adaptive strategy (3.4)**
>
>     The parameter $\theta_k$ is an important parameter to control the quality of SSN direction $\Delta w_k$. When $\theta_k$ is large, $\Delta w_k$ usually leads to a slow yet stable convergence. When $\theta_k$ is small, $\Delta w_k$ can be a bad SSN direction but the rate of convergence will be fast if $\Delta w_k$ is good. If $\frac{\rho_k}{\||\Delta w_k\||^2}$ is small, $\Delta w_k$ is usually a bad SSN direction and we increase $\theta_k$. Otherwise, we decrease it.
>
> 5. **In the proof of Theorem 3.4, the auxiliary sequence $v_k$ is not considered. It seems that the strategy in line 238 cannot be neglected and $w_{k+1} = v_{k+1}$ needs to be precluded under the conditions of Theorem 3.4**.
>
>     Thank you for your insightful comments! Let us clarify why the current theoretical analysis does not need to preclude the case $w_{k+1} = v_{k+1}$ under the conditions of Theorem 3.4. The key ingredient is that we have assumed that the iterate $w_0$ is *sufficiently* close to one nondegenerate optimal solution. Then one regularized SSN step is guaranteed to achieve a quadratic rate. Since one EG step achieves a linear rate (it is a first-order method), we know that one regularized SSN step reduces the residue norm more than the EG step under the conditions of Theorem 3.4 (i.e., sufficiently close). This implies that $w_{k+1} = v_{k+1}$ can be precluded given that the iterate $w_0$ is *sufficiently* close to one nondegenerate optimal solution.
>
> We thank you again for your detailed reading and your constructive input!

---

### Author Rebuttal · Authors · 2023-08-09

**We would like to thank PCs, SACs, ACs and the reviewers for their efforts on evaluating our paper.** We appreciate that the reviewers pointed out the importance of the problem and of our algorithm given the increasing popularity of computational OT. All the comments will be addressed in the revised version of our paper, and we will release the source code if the paper is accepted.

Besides the specific responses that we have provided to each reviewer, we follow the suggestion of Reviewer bQMP by comparing our method with the pure extragradient (EG) method. The preliminary results show that our method consistently outperforms the pure EG method and can output a high-accurate solution in terms of the residue norm. The experimental setup here is the same as that used in the main paper: we fix the dimension $d=5$ and vary the sample size $n \in \{50, 100, 200, 300, 400, 500\}$. For the pure EG method, we tune the stepsize and set it as $0.01$.

---

### Decision · Program_Chairs · 2023-09-21

**Decision:**

Reject

**Comment:**

After carefully going through all the reviews, rebuttal, and the discussions, and going over the paper, I have come to the following conclusion.

-- The technical contributions of the paper are solid and the problematic is significant and timely. However, there are several concerns raised by reviewers, especially reviewer 7do7, which are mainly based on the presentation and clarity of the contributions. While I agree with the authors that some of the comments by reviewer jxBx are vague (which I did not put too much emphasis), I agree with them on the overselling part of the contributions. Unlike reviewer 7do7, I do not believe "just trusting" the authors for implementing all the mentioned changes is sufficient. With all the suggested changes implemented, the paper would require another round of revision, hence I consider the score of reviewer 7do7 as a weak reject, as they increased their score based on assuming all the changes will be implemented.

From this perspective, I do believe that the paper would greatly benefit from another round of revision. Hence I am recommending a borderline rejection.